# Circular Economy and Environmental Sustainability: A Policy Coherence Analysis of Current Italian Subsidies

Andrea Molocchi

Ricerca sul Sistema Energetico—RSE SpA, 20134 Milano, Italy; andrea.molocchi@rse-web.it; Tel.: +39-3891903736

**Abstract:** Policy instruments for a circular economy and for environmental protection both aim at sustainable development, but do not necessarily share the same goals. The purpose of the paper is to assess the coherence with the EU-recognized circular economy principles of a large set of subsidies currently in force in Italy: those listed in the Italian Catalogue of Environmentally Harmful Subsidies and Environmentally Friendly Subsidies. The method is inspired by the OECD's "Checklist" method and "Policy Coherence for Sustainable Development" approach recommended by the UN 2030 Agenda, which can be usefully applied to all policy instruments, including subsidies and green budgeting. The analysis allows one to identify as many as 56 potentially harmful subsidies for the circular economy in Italy, for a financial value of at least 13.5 billion euros in 2019, and 75 potentially friendly subsidies for the circular economy, for at least 13.0 billion euros. A total of 25% of subsidy schemes analyzed with a circular economy perspective have contradictory effects when compared to the Catalogue's results obtained with an environmental perspective. The results of the study underline the importance of coordination between environmental and circular economy policies in achieving sustainable development goals. The analysis can be considered a "pilot study" on a national case; its method can be easily replicated by administrations also in ex-ante evaluation of new subsidy schemes.

**Keywords:** sustainable development policies; subsidies; circular economy; environmental fiscal reform; green budget; policy coherence; taxonomy





## 1. Introduction

This paper deals with the policy coherence of a nation's subsidies with circular economy (CE) principles in the context of environmental sustainability policies [1,2]. Subsidies are a category of economic instruments for environmental policies, alongside other types of economic instruments such as taxes, tradable permits, regulated tariffs, deposit-refund schemes, road pricing [3,4]. Economic instruments contribute to the mix of policy instruments, with command and control regulatory instruments, voluntary agreements, and information-based instruments [5]. In a CE policy framework, subsidies are used to overcome the financial and economic barriers to the investments needed to increase the recycling capacity of the waste industry to support expensive resource efficiency projects and to promote the organizational and behavioral changes in production and consumption [6–8].

According to the Global Material Resources Outlook [9] produced by the OECD (Organization for Economic Co-operation and Development), the global primary materials demand needed to sustain future economic growth is projected to almost double in the next 40 years, increasing from 89 giga-tons (Gt) in 2017 to 167 Gt in 2060, despite the expected savings allowed by innovation and structural change. This scenario will be coupled with a wide range of environmental consequences, including acidification, climate change, eutrophication, land use, as well as water, human and terrestrial ecotoxicity, representing an important challenge for environmental sustainability. According to the Outlook, CE inspired policies are needed to minimize environmental impacts: "The per-kg

environmental impacts of secondary materials are estimated to be an order of magnitude lower than those of primary materials. Policies that further ramp up the transition to secondary materials use and promote circularity will thus lead to overall reductions in environmental impacts." [9] (p. 16) Depletion of non-renewable resources is also at stake. In many countries and particularly in the EU, there is a growing concern that many green technologies needed to move towards a low-carbon economy will imply a considerable increase in the demand for *critical raw materials* (materials that are most important economically and have a high supply risk [10]) and that a higher policy effort to increase the rates and quality standards of recycling processes is needed to reduce dependence from primary materials supply [11]. For example, new efficient solar photovoltaic cells may contain critical materials that could generate bottlenecks in the future, such as tellurium, copper, silver, cadmium, indium or gallium [12,13]. To overcome these constraints, policy measures aimed at improving recycling rates of critical materials from 0.1% to 4.6% per year are needed to avoid material shortages or restrictions in green technologies [12]. These challenges also depend on the availability of sufficient market and public incentives for the recycling industry to overcome the phenomenon of "downcycling" that obstacles the supply of recycled critical materials ("downcycling" happens when the value obtained from a recycled product is lower than the value of the materials contained in the original product, due to technical feasibility constraints and high costs of optimal recycling [14]). The recovery of critical raw materials and avoidance of downcycling are future challenges for a CE paradigm that needs adequate financial support both at the R&D and commercial scale.

Governments will need a lot of financial resources to promote the investments and changes needed to the transition to a CE. But this is just one side of the issue, since there is another role played by subsidies in the perspective proposed by this paper: precious public resources currently spent for many other economic and social reasons could push towards the conservation of current resource wasteful practices, slowing down the efforts towards CE.

Are subsidies in a country favoring or hindering the transition to the CE? This seems to be a rather new question both in the academic literature and in public policy analysis. The issue of policy coherence of subsidies has been much more debated in relation to environmental objectives (are a nation's subsidies harmful or favorable for the environment?), at least for fifteen years (see literature review in Section 2). The set of environmental objectives enshrined in countries' laws and regulations, such as reducing air pollution, waste prevention, recycling and disposal, protecting natural ecosystems, is somehow overlapping and partly different as compared to the set of objectives of the CE perspective. Contradictions between the two perspectives can be easily found (for example, in subsidies to substitute still functioning low energy efficient devices with higher standard ones, without a proper recovery and recycling scheme). Coordination within and between the two perspectives must be searched. To coherently align a nation's subsidies with the environmental and CE objectives under the auspices of the UN policy framework of the 2030 Agenda for sustainable development [2], the research community should start to detect contradictions of subsidies with policy objectives, by developing methods to do this. That's the why this paper also deals with the differences and commonalities between the CE and the environmental perspectives when analyzing subsidies.

One of the difficulties encountered in assessing the policy coherence of subsidies with CE is the complex and constantly evolving nature of the CE paradigm (see Appendix A—CE and its concepts: an historical overview). The choice of this paper is to anchor analysis to the CE concepts and principles developed by the EU policy. An official definition of CE is now available—recently provided by the Taxonomy for sustainable finance (Regulation EU 2020/852 [15]), that included CE in the list of the six environmental objectives that may be reached to obtain the qualification of environmentally sustainable activity (1) climate change mitigation; (2) climate change adaptation; (3) the sustainable use and protection of water and marine resources; (4) the transition to a CE; (5) pollution prevention and

control; (6) the protection and restoration of biodiversity and ecosystems) [15] (art. 9). The availability of an official reference on CE is an opportunity for Member States to start assessing the alignment of existing subsidy schemes with the CE and the other environmental objectives of the EU policies.

Are the CE principles related to the EU definition fully integrated into national public subsidies? Can a subsidy favoring CE be, at the same time, harmful from an environmental point of view? The purpose of the paper is to assess the policy coherence with the EU recognized CE principles of a large set of environmentally significant subsidies (potentially harmful or friendly for the environment) currently in force in Italian legislation, with the aim of providing to policy-makers an informative basis on subsidies that are potentially harmful or friendly for the CE, to increase integration of CE into environmental policies. The paper also compares the different outcomes obtained when the two perspectives are separately applied in the evaluation of subsidies, with the purpose of clarifying overlapping or autonomous areas of intervention, which, if not properly coordinated, can cause contradictions between them, reducing the effectiveness of their contribution to the overarching sustainable development aim. The paper will achieve these goals by finding out a robust set of CE principles to be used in the analysis of subsidies (examination of the EU Taxonomy regulation and other policy documents) and by applying a simple and practical method to assess subsidies against these principles [16,17].

The choice of the Italian case for analyzing subsidies is mainly due to the availability in Italy of an official Catalogue of Environmentally Harmful or Environmentally Friendly subsidies (from now on "the Catalogue") [18], which has now arrived at the third edition [19]. This document offers a wide set of subsidies (more than 170), each one with its "subsidy identity card" that synthesizes the regulatory references of the subsidy, allowing the readers to access the legal sources and develop their own opinion or analysis: it is a sufficiently transparent source on which to build the analysis of policy coherence of a country subsidies with the EU CE principles.

The suggested approach can be easily replicated by national administrations in the ex-ante and ex-post evaluation of subsidy schemes to find out friendly or harmful cases, offering to policy-makers the informative basis to enhance the effectiveness of CE policies in Italy and other countries. The approach can also be considered for green budgeting reviews (green budgeting means using the tools for state budget policy-making to help achieve environmental goals; this includes evaluating environmental impacts of budgetary and fiscal policies and assessing their coherence towards the delivery of national and international commitments [20]). It must be underlined that analysis of subsidies provided by this paper does not pretend to suggest how to reform the identified harmful subsidies, nor does it intend to make proposals on the optimal mix of taxes or subsidies for the green economy in Italy.

Regarding the outline of the paper, Section 2 reviews the available academic literature and policy reports on the main issues raised by this research. Section 3 is dedicated to present the materials and methods used in the analysis, firstly by describing the subsidy data set (the Italian Catalogue, Section 3.1), followed by a section presenting the EU regulatory references for the CE principles applied in the analysis (Section 3.2) and a third section describing the methods used to assess subsidies against the CE principles (Section 3.3). Section 4 is devoted to present the results of the analysis. Subsidies that have been classified harmful for the CE are the first to be commented (Section 4.1), followed by those that have been classified friendly for the CE (Section 4.2). In a third section (Section 4.3), the results obtained by the policy coherence analysis of subsidies through a CE perspective are compared with those obtained through a strict environmental perspective (Catalogue results). In Section 5 the limitations of the approach are discussed, and in the final Section 6, the main conclusions from the analysis are drawn.

## 2. Literature Review

In general terms, the paper can be placed in the broad and long-dated knowledge area of economic instruments for environmental policy, particularly at the border of those contributes assessing environmentally harmful subsidies or devoted to environmental fiscal reform [16,21–23]. To our knowledge, this is the first attempt to analyze the coherence of a country subsidies with the CE principles in a sustainable development policy framework. Since the paper shifts the conventional environmental perspective to adopt a CE one, it opens new links with research related to CE policies, particularly with the area of policy mixes for the CE [24–28] and with the sub-area of economic instruments for the CE. The main contributions in the latter sub-area currently concern the potential of environmental economics in favoring the CE [29], taxation of non-renewable resources [30,31], optimal taxation frameworks for CE [32] and designing of direct subsidies to overcome investment barriers to eco-innovation [6,7]: these are all bordering but different areas from the issue of policy coherence of subsidies raised by this paper. The literature context of the paper is traced in the policy-oriented research streams on fossil fuel subsidies (FFS) and environmentally harmful subsidies (EHS), on which the current chapter will focus on. Appendix A provides to the reader a historical overview of the main CE concepts and "schools of thought" and their uptake in policies (particularly the EU one), offering further insights into the literature context of the paper.

When a new policy paradigm emerges, such as the EU environmental policies in the seventies and eighties [33] and the more recent affirmation of the CE policy in the EU [15,34,35] (Appendix A), it is important to analyze existing subsidies in a country legislation in order to check their coherence with the new policy objectives, preventing unwished outcomes and waste of public financial resources. Oosterhuis and ten Brink [21], who dedicated the introductory section of their book on EHS to reconstruct the birth of the approach of assessing EHS, say that the pioneers were economists at the World Bank in the early nineties, exploring the relationship between energy subsidies (provided in early days to promote economic development) and their adverse effects on the emerging issue of climate change. The study and disclosure initiatives taken by environmental think tanks and non-governmental organizations in the years between the 1992 Rio Conference and the 1997 Kyoto Protocol increased the social and policy awareness on the amounts of public financial flows directed to fossil fuel-based industries, highlighting their contradiction with the emerging climate mitigation policies [21]. In the following years, the OECD, that already had a tradition in reporting on agricultural, fisheries and energy subsidies, played an essential role in transforming the initial concept of FFS into a more general approach to analyze and reform EHS in the main sectors of activity and in relation to all relevant environmental impact factors [16,36,37]. Besides OECD work, case studies and national inventories of potentially EHS, most of them focusing on energy subsidies, were made for Australia, Germany, Poland and the EU [21]. Particularly important are the two reports made by the Institute for European Environmental Policy (IEEP) for the European Commission in 2007 [38] and 2009 [22]: building on the OECD work on EHS, they reviewed the available definitions of subsidies, allowed a refinement of methodologies to detect and assess EHS, developed techniques to quantify their financial size and also provided guidelines for governments to analyze options for EHS reform or removal. According to Oosterhuis and ten Brink [21], two phases of the analysis must be clearly separated: in a preliminary phase, subsidies in a country must be "reconstructed" (their legal sources are often scattered and "hidden" in various regulatory streams) and assessed from an environmental point of view; in the second one "EHS should be disclosed, debated and, where appropriate, reformed, taking into account the pros and cons of specific reform option. ( . . . ) Whether an EHS needs reform will require the wider picture of environmental, social and economic benefits and trade-offs to be understood" [21] (p. 8).

One of the main points of discussion in the literature on EHS is the lack of consensus on the definition of subsidies to be considered in the analysis of their environmental harmfulness [21,39]. IEEP [22,38] and other more recent European reports [40,41] adopt

broad definitions inspired by the OECD's [16] ("subsidy is a result of a government action that confers an advantage on consumers or producers, in order to supplement their income or lower their costs"), that allow several support measures to be considered as subsidies, such as directs transfers of funds, market price support, covering liabilities, feed in tariffs, tax exemptions or rebates. The International Energy Agency (IEA), that has been measuring fossil-fuel subsidies world-wide for more than a decade (its work is incorporated in major publications of the World Energy Outlook series [42]), considers as subsidies those policy measures that lower the price of fossil fuel products below their international market price thus making them more affordable for beneficiaries (the "price gap approach" [43]). Both the OECD and IEA definitions do not include implicit subsidies that result from the non-internalization of externalities and lack of social cost pricing. This approach is currently promoted by the IMF (International Monetary Fund) [44–46] and accounts for both of pre-tax and post-tax consumer subsidies. Pre-tax subsidies exist when energy consumers pay prices that are below the costs incurred to supply them with this energy. Post-tax consumer subsidies exist if consumer prices for energy are below supply costs plus the efficient levels of taxation. The efficient level of taxation includes two components. First, energy should be taxed the same way as any other consumer product. Second, some energy products contribute to local pollution, traffic congestion, accidents and global warming. Efficient taxation requires that the price of fossil fuels should reflect the external costs that fall on society [46]. In most countries, taxes on energy fall far short of the efficient levels covering the external costs [46]. Exclusion or inclusion of externalities in the subsidy estimate can account for a variance from $500 billion to more than $5 trillion in valuation of global fossil fuel subsidies [47].

Notwithstanding differences in definitions, besides the efforts of international institutions including the OECD, IEA, IMF, several non-governmental organizations have become active researchers of FFS and EHS with the aim to promote subsidy reforms at the country level in support of climate change mitigation and sustainable development [39,48]. These contributions are often advanced in the wider framework of an environmental fiscal reform aimed to shift tax pressure from labor to environmental impacts and resources [23]. Unfortunately, until now, little attention was paid by these proposals to the inclusion of the emerging policy paradigm of the CE in the needed subsidy and fiscal reform, under an integrated approach.

The main success of FFS reform initiatives is the inclusion in the 2015 UN 2030 Agenda for Sustainable Development of a specific target (12.c) for reforming FFS [2] (p. 23). A few months later, in the Paris Conference of the Parties (COP) 21 climate negotiations under the United Nation Framework Convention on Climate Change (UNFCCC), the international community hasn't managed to confirm the same commitment in a binding manner. However, even if the Paris agreement makes no explicit reference to reform FFS, it provides a voluntary framework based on Nationally Determined Contributions (NDCs), which is currently used by many countries to engage in reforms of FFS [39,49].

The need for a reform of FFS has been stressed several times in the G7 and G20 Summits, starting from Pittsburgh—US (2009) [50], where G20 committed to "To phase out and rationalize over the medium-term inefficient fossil fuel subsidies while providing targeted support for the poorest". In 2015 the G20 launched a voluntary peer-review program of national reports on fossil fuel subsidies. The exercise has involved China and the USA in 2016, Mexico and Germany in 2017, Italy and Indonesia in 2018 [51]. The Self-report of Italy [52] and the Peer review conducted by an evaluation team made up of international experts from the G20 [53] were published by the OECD in April 2019 [51]. A parallel voluntary peer-review process has been activated by Asian Pacific Economic Cooperation (APEC) countries since 2014 [54]. About thirty Governments (mostly APEC) are actively pursuing reforms of specific subsidies, of which 13 countries, including China, explicitly integrate FFS reform into their NDCs [39].

At the G7 summit in Ise-Shima—Japan (2016) [55], a commitment to phase out FFS was signed with the inclusion of a deadline in 2025, inviting all other countries to follow

the example. In the G7 Environment of 12–13 June 2017, held in Bologna under the Italian presidency, the G7 countries reaffirmed the commitment of the previous year [56] (p. 10). Again, even if the final text has the merit to include a paragraph on CE and an Annex on resource efficiency (5-year Bologna Roadmap), there is not any analogous commitment for the integration of the CE concepts in the needed environmental fiscal reform or statements to increase policy coherence of taxes and subsidies with CE principles, for pursuing the UN sustainability goals.

Although many G7 and G20 States are part of the EU, at European level the commitment to phase-out FFS has never been translated into a community-wide legislative act, due to the conflicting positions and the substantial autonomy of the Member States in fiscal policies. The issue of phasing out fossil fuel subsidies has only recently become part of the EU's agenda. The European Green Deal program of the von der Leyen Commission mentions many times the goal of removing subsidies to fossil fuels (see [57] (pp. 10, 17–18, 21). One of the main commitments is the revision of the ETD (Energy Tax Directive, 2003/96/EC), which regulates the exemptions and allowances of excise duties on energy products that are decided by the Member States. According to the OECD broad definition [16], tax expenditures are indirect subsidies. The Commission is expected to publish the text of the Directive proposal by mid-2021 [58].

The analysis of the main EU policy documents on CE, including the last EU Action Plan for the CE [35], that is made in Supplementary Material 1 (SM 1—Analysis of CE concepts and principles in the main European policy documents on CE), highlights a limited vision on the role of economic instruments, subsidies included, in the needed transition. While recognizing the relevance of direct incentives for CE investments and of specific taxes as the tax on landfill waste disposal, both the old [34] and the new version [35] of the European Commission's CE Action Plan do not mention the potential of a coherent regime of subsidies and taxes on resource use to meet the CE needs [32]. For example, notwithstanding the recent Commission commitment to reduce fossil fuel subsidies in the European Green Deal [57], the new version of the CE Action Plan [35] doesn't mention any commitment to at least start finding out existing subsidies "hidden" in national legislations that favor the conservation of "non-circular" and resource wasteful approaches. The EU Commission is willing to explore the potential of a broader and less penalizing approach based on "green budgeting" [20]. The EU Green Deal announced a cooperative work plan towards green budgeting rules: "National budgets play a key role in the transition. Greater use of green budgeting tools will help to redirect public investment, consumption and taxation to green priorities and away from harmful subsidies. The Commission will work with the Member States to screen and benchmark green budgeting practices." [57] (p. 17). This is a promising stream that could include in its approach potentially harmful subsidies for CE, as well as FFSs and EHSs.

Coming to the national level of policies, country studies and reports inspired by the broader concept of EHS subsidies are not missing, particularly in the EU (reviews are provided by [21,54,59]), but it's difficult to find similar works applied to the CE.

Among the official national reports, in Germany, the Federal Environment Agency (Umweltbundesamt—UBA) regularly publishes since 2008 a report on EHS [41]. The last edition (2016) [60] covers the following sectors: energy; agriculture, forestry and fisheries; transport; construction and housing. The amount of EHS is estimated for 2014 at € 57 billion.

In France, the 2012 "Sainteny report" for the Prime Minister [61] probably represents the first attempt to assess a nation's subsidies under the specific perspective of biodiversity impacts by applying a DPSIR approach (Driving force-Pressure-State-Impact-Response). Subsidies have been analyzed for their influence on the following pressures and driving forces: habitat destruction and degradation; over-exploitation of natural resources; air emissions (including greenhouse gases), soil and water pollution; introduction and proliferation of alien species. Unfortunately, this promising approach to the evaluation of subsidies hasn't been further developed with follow-up reports. In September 2020, the

French government released a further innovative report [62], produced under the OECD Paris collaborative on green budget [20], that maps the environmentally positive or harmful impacts of France 2021 state budget proposal. The main aim of the report is to "assess the compatibility of the state budget with the international commitments of the France Government, in particular the Paris Agreement" [62] (p. 6). Even if this is not a report dedicated to subsidies, since it enlarges the analysis to any form of public expenditure (including current public expenditures for health, education, and also including tax expenditures), it represents a promising even if challenging approach for the application of environmental assessment methods to a state budget (for a first critical analysis of challenges raised by this approach, see [63]).

In Italy, the first step towards a reform of EHS was made with the introduction of the "national catalogue of environmentally harmful (EHS) or environmentally friendly subsidies (EFS)" (art. 68 of Law 221/2015 [18]). The Catalogue is an information tool conceived to facilitate the political discussion on environmental fiscal reform. It reviews and identifies EHS and EFS through the use of "subsidy identity cards", as recommended by IEEP reports [22,38]. The elaboration of the Catalogue enabled the Italian Government to discuss and evaluate the complex issue of reforming FFS: a commitment for a reform has been included in the Italian Integrated National Energy and Climate Plan (INECP), discussed with the EU Commission during 2019 and definitively approved on 17 January 2020 [64]. INECP introduces for the first time in an Italian energy policy document a plan to reform fossil fuel subsidies. It provides for a gradual approach according to three priority levels (see Tables 62–64 in [64], pp. 286–293). In addition, the Italian 2020 budget law has established an inter-ministerial Commission for the study and drafting of policy proposals for the ecological transition and the reduction of environmentally harmful subsidies [65].

## 3. Materials and Methods

### 3.1. The Data Set Used

This paper adopts the same subsidies (and the underlying assumptions on their definition) analyzed by the third edition of the Italian Catalogue [19] (those in force until the end of 2018) as the informative basis to fulfil its aim (analysis of the policy coherence of a country subsidies with the CE principles). More recent subsidies, some of which might be relevant for CE, are left outside the scope of the paper in order to ensure uniformity in the comparison between the environmental assessment of subsidies made by the Catalogue and the analysis of coherence with the CE (Section 4.3).

Here follows a summary of the Catalogue, focusing particularly on the following aspects:

- aims and definitions;
- methods of environmental assessment;
- references to emerging CE policies and measures.

The main results of the Catalogue [19] are summarized in Appendix B.

### 3.1.1. Aims and Definitions

According to its institutive law, the aim of the Catalogue is to make an updated list of environmentally harmful or friendly subsidies to support the Government in the implementation of a series of international and EU commitments and recommendations on environmental policy. In practice, the Catalogue is a report that reviews all existing subsidies in force in Italian Legislation, which could have a positive or negative environmental impact, with the purpose of classifying them as EHS or EFS, through a subsidy "identity card" that summarizes the main features of each subsidy and explains the reasons of the classification. Basically, it carries out an *ex-post* assessment of the most significant subsidies from an environmental point of view.

It is not the aim of the Catalogue to plan a reform of EHS or to advance proposals for removing specific subsidies. It is only an information tool at the service of decision-makers (Parliament, Government, etc.), which is also open to comments and contributions from

the scientific community (each edition of the Catalogue undergoes a consultation phase, that is open to all stakeholders).

For the provisions of art. 68 of Law 221/2015, subsidies "are considered in their broadest definition and include, among others, incentives, tax concessions and exemptions, subsidized loans directly aimed at protecting the environment" [18]. This explains why the Catalogue examines not only direct transfers but also "indirect" subsidies, such as tax expenditures, implicit subsidies and cross subsidies paid through regulated tariffs on public services (electricity, gas, waste and water services). On the other hand, subsidies intended as the monetary value of environmental damages not properly transferred into market prices by the use of environmental taxes (optimal tax revenue loss) have so far been excluded from the Catalogue, even if there is a specific strand of literature providing methods and post-tax external costs estimations under this approach [44,45,66].

### 3.1.2. Environmental Assessment Methods of the Catalogue

Law 221/2015 does not require specific environmental method or criteria to follow in the subsidies' classification into EHS or EFS. The Catalogue adopts an evaluation process of subsidies "freely" inspired by the OECD's "quick-scan" [37], "checklist" [16] and IEEP's [22] methods, that aims to verify the "qualitative compliance" of the main effects of the subsidy with the environmental principles and objectives "incorporated" in current legislation. When the environmental evaluation brings contrasting outcomes—positive for certain environmental impacts and negative for others- and further assessment is needed, the subsidy is classified as Uncertain (EUS). The Catalogue method is summarized in [19] (pp. 24–25, 129–131); further information is provided in [67].

One of the limits is that the environmental criteria applied in the evaluation are made explicit in the subsidy identity card according to the cases. Transparency can be improved by systematically adding a label in the subsidy identity card quoting the subsidies' environmental impact types (the environmental objectives impaired or favored). During this assessment process, authors applied a mix of evaluation methods, such as:

- literature references (academic studies, research reports, literature reviews, environmental assessment manuals, guidelines for evaluating external costs, etc.);
- environmental indicators provided by statistical institutes;
- regulatory references (in some cases, the assessment refers to community laws that classify certain energy products as "renewable sources" [68] or "alternative fuels" [69], providing the legal basis for their financial support).

It is important to underline that the Catalogue doesn't rely on a systematic evaluation method based on a common metric of environmental impacts (it does not quantify the overall environmental impact or benefit of each subsidy). Although often supported by indicators and evidence from the literature, the assessment is fundamentally based on checking the coherence of the effects of the subsidy in relation to a set of principles and targets that are not systematically made explicit. In other words, the classification of the Catalogue remains a "structured" judgement.

### 3.1.3. References to Emerging CE Policies and Measures

The legislative mandate of the Catalogue is to apply an environmental perspective in assessing subsidies, excluding other assessment criteria, such as economic or social effects. Even if the CE shares some commonalities with the environmental perspective (for example, efficient use of non-renewable scarce natural resources), it is also a policy paradigm introducing different economic and industrial priorities: we should not expect from the past editions of the Catalogue a systematic analysis of the compatibility of current subsidies in Italy with the EU principles of the CE, which is indeed the aim of this work. The latest edition of the Catalogue [19] started to note the increasing relevance of CE both in the EU [34] and National policies [70,71] and classifies new dedicated incentives scattered in recent financial laws. The Catalogue has classified them as EFS (these subsidies are examined from the point of view of CE in Section 4.2.4).

*3.2. CE Principles for Subsidy Analysis*

In order to assess the policy coherence of subsidies with CE, the main EU policy documents on CE have been reviewed to find out a robust reference on CE principles. In particular, the following documents have been considered:

- the first EU Action Plan for a CE (2015) [34];
- the 2017 Commission guidelines on the role of waste-to-energy in the CE [72];
- the new EU Action Plan for a CE (March 2020) [35];
- the EU Regulation n. 852 of 18 June 2020 on the Taxonomy for sustainable finance [15], that provides an official definition and a list of principles of the CE;
- the categorization system for the CE developed by the Circular Economy Finance Expert Group (CEFEG), an expert group supporting the Commission on CE, published in March 2020 [73].

The discussion of the above-mentioned documents is provided in SM 1—Analysis of CE concepts and principles in the main European policy documents on CE. The main conclusions of the analysis of the CE principles mentioned by such documents are summarized below.

For the aim of this paper, the CE principles established by art. 13 of the EU Regulation 2020/852 [15] have been adopted. Even if comprehensive, this list of principles may appear generic in relation to specific natural resource types, leaving space for interpretation in some cases of subsidies, particularly those related to energy recovery from waste streams, agriculture and real estate/building.

To improve analysis of subsidies, principles listed by art. 13 of the Taxonomy have been integrated by more detailed guidelines, as the CEFEG categorization system of CE [73]. This system has the advantage of making explicit CE principles related to two natural resources generally neglected in the reflections on the CE:

- land (CEFEG circular category "2d"): artificially degraded land and brownfield sites are not seen as "definitively consumed land"; on the contrary they are seen as resources that can be reused, after a proper environmental reclamation and restoration;
- water (CEFEG circular category "3d"): water used and subsequently discharged (wastewater) is also seen as a resource that—especially in situations of scarcity—can be usefully treated and distributed for new uses.

However, in contradiction with this "open view" on soil and water recovery, CEFEG's proposal is highly restrictive with regard to energy recovery, contrasting with art. 13 of the EU Regulation 2020/852 and with the official position of the European Commission on the role of waste-to energy in the CE [72]. Energy as a natural resource is included in point (a) of art. 13 [15], as is the recovery of energy from waste streams provided that it respects the other priorities of the EU waste hierarchy (point j of art. 13). The Commission's guidelines on the role of waste to energy in the CE [72] include different types of energy recovery from waste into CE models, according to a prioritization consistent with the EU waste hierarchy (from top to bottom): production of biogas from anaerobic digestion of organic waste; co-combustion of waste for the production of cement and lime; co-combustion of waste for the production of electricity in power plants; incineration of waste with high standards of energy recovery; incineration of waste with low standards of energy recovery or without energy recovery; energy utilization of captured landfill gas. For example, according to the Commission guidelines [72], public subsidies for investments in waste incineration plants with energy recovery can be granted only in limited and well-justified cases, when the plant meets high energy efficiency requirements, there is no risk of overcapacity, and the objectives of the waste hierarchy are fully respected. Since some of the Italian subsidies are related to waste management and energy recovery, the Commission's guidelines have been adopted as a third reference for CE principles.

Summary tables and details on the CE principles mentioned by these three references are provided in SM 1—Analysis of CE concepts and principles in the main European policy documents on CE.

### 3.3. The Assessment Method

The method of policy coherence analysis applied by the paper is a simple and practical method, similar to the one applied by the Italian Catalogue, that is freely inspired by the "Checklist" method to assess EHS [16] and the OECD's approach "Policy Coherence for Sustainable Development (PCSD)" [17]. PCSD is an approach and policy tool developed by the OECD to systematically integrate the economic, social and environmental dimensions of sustainable development at all stages of domestic and international policy-making. The method is linked with the practical implementation of the 2030 Agenda [2] and its 17 Sustainable Development Goals (SDGs). PCSD has been embodied in SDG n. 17 ("Means of implementation for all SDGs") as a specific target (17.4), with the aim to support Governments with a method to increase coherence between policy measures aimed at the 17 SDGs and achieve the Agenda's primary aspiration of "shifting the world onto a sustainable path".

The assessment method is based on a qualitative evaluation procedure based on a series of steps and on a checklist that is, in this case, the list of CE principles. The evaluation aims to the classification of each subsidy scheme under scrutiny into three classes that represent the final outcome (potentially positive or negative for the CE) or the need for further analysis (uncertain for the CE). The analysis of each subsidy begins with the reconstruction of the legislative and regulatory framework that states the aim and the features of each subsidy. On the basis of the EU CE principles listed in the three regulatory sources previously mentioned [15,72,73], the direct effects of the subsidy are examined against each principle to find out the sign of each effect. If the effects positively contribute to at least one CE principle and do not harm any other principle, the subsidy is classified as friendly for the CE; if the effects contradict at least one CE principle and do not contribute (are not relevant) to other CE principles, the subsidy is classified as harmful for the CE. If the effects are friendly for a principle and at the same time harmful for any other principle (or the effects are contradictory for a certain principle), and uncertainty cannot be reduced by the use of estimates, results of case studies or indicators, the subsidy is classified as uncertain for the CE.

The policy coherence analysis has actually led to a reclassification of the Catalogue subsidies according to a formally broader template which, in addition to the Catalogue's three classes (harmful, friendly or uncertain), provides for two further classes, for a total of five classes:

- **potentially harmful subsidies for the circular economy (HCE):** subsidies with at least one CE principle that is harmed by the effect of the subsidy, all other principles remaining substantially neutral. For example, a tax discount on Liquified Petroleum Gas (LPG) used for water heating (see Section 3.1.1) favors higher energy consumptions through energy price reduction and distorts competition with solar thermal; this is harming the CE principles of efficient use of scarce natural resources (fossil fuels) and of renewable source use, while other CE principles are not favored;
- **potentially friendly subsidies for the circular economy (FCE):** these are subsidies with at least one CE principle that is favored by the effect of the subsidy, all other principles remaining substantially neutral. For example, the subsidy for the maintenance of olive tree plantations (AR.SD.08 in Section 4.2.3) increases the lifetime of plantations and prevents land consumption (artificial coverage), with no significant adverse effects on other CE principles;
- **uncertain subsidies for the circular economy (UCE):** subsidies for which the comparison with the CE principles has led to contrasting signs of the effect (for example, see the case of the lower excise duty on diesel as compared to petrol, Section 4.1.1);
- **neutral subsidies for the circular economy (NCE):** subsidies that have no significant effects on the CE principles. This category has been added to prepare the subsequent comparison of results with the Catalogue. In fact, the category of neutral subsidies from the environmental point of view is implicit in the evaluation methodology of the Catalogue [19] (p. 129). The Catalogue's assessment procedure starts with the

preliminary selection of those existing subsidies that are potentially relevant for the environment, that are subsequently submitted for evaluation and classification within the Catalogue files;

- **subsidies not anymore in force in 2018**. This class is due to the fact that during the CE analysis emerged that 5 subsidies included in the 3rd edition of the Catalogue (which should cover subsidies in force in Italian legislation until the end of 2018) ended in 2017 and were not anymore in force during 2018 (they should have been excluded in the yearly updating of Catalogue). These subsidies have been excluded from the circularity assessment.

Similarly to the Catalogue, the evaluation and final classification of the subsidy can be supported by various instruments, such as regulatory analysis of the waste management provisions on the end-of-life of subsidized products (see for example the case of subsidies to PV panels, Section 4.2.1), or sectoral reports on recycling perspectives (see the case of electric vehicles batteries, TR.SD.06 Section 4.2.2), or by life-cycle assessment (LCA) studies and indicators, when available.

Circularity analysis has taken into consideration all the subsidies examined by the Catalogue, in order to capture and count all possible sign confirmations or variations of the circularity assessment as compared to the environmental assessment. The base of the analysis is the "subsidy identity card" provided by the Catalogue for each subsidy. In our analysis the reference to the same code of the subsidy identity card adopted by the Catalogue has been maintained. The code allocates subsidies into 5 sectors (Energy—EN, Transport—TR, Agriculture and fishing—AP, Other subsidies—AL, VAT concessions—IVA) and two categories (direct subsidies—SD, indirect subsidies—SI), for example the code TR.SI.04 means "subsidy n. 4 of the list of indirect subsidies of the transport sector". The analysis considered in particular the "motivation" field of the subsidy identity card, which explains the environmental reasons that led to a certain classification (EHS, EFS or EUS). In fact, some reasons can also fall within the principles of circularity (typically those related to a greater resource efficiency, energy efficiency included). One of the main differences with the Catalogue approach is that in this analysis, all principles used are made explicit (for reasons of synthesis they are reported in the respective legal sources in SM 1) and those CE principles that allowed to successfully qualify a subsidy are traced in SM 2—*Tables of Harmful subsidies for the circular economy (HCE) and Friendly subsidies for the circular economy (FCE)*.

## 4. Results

This Chapter illustrates the results of the policy coherence analysis of the environmentally relevant subsidies in Italy with the EU CE principles. Sections 4.1 and 4.2 comment the detailed results the outcome of which is respectively Harmful for Circular Economy (HCE) or Friendly for Circular Economy (FCE). Given the large number of subsidies examined, the presentation of results is sorted by sector, following the same classification of the Catalogue (e.g., 4.1.1 Energy, 4.1.2 Transport, 4.1.3 Agriculture and fishing, 4.1.4 Other subsidies, 4.1.5 VAT allowances). Overview tables that report the outcome of the policy coherence analysis for each subsidy and sector are provided in SM 2. These tables show for each subsidy:

- the Catalogue code (sector, type, numbering) and subsidy's title;
- the environmental qualification of the Catalogue (EHS, EFS or EUS);
- the CE principles that respectively motivate the new classification of subsidies as harmful for the circular economy (HCE) or Friendly (FCE);
- the financial value of the subsidy in the period 2017–2019 (as said in Section 3.2, not all subsidies have been financially quantified by the Catalogue; the aggregated values should be considered as minimum values).

In Section 4.3 the summary results of the circularity analysis (in terms of number of subsidies) are compared with those of the environmental analysis of the Catalogue, focusing in particular on the commonalities and differences of the outcome of the two evaluation approaches.

### 4.1. Subsidies That Are Harmful for the Circular Economy (HCE)

The analysis of coherence with circularity principles identified 56 potentially harmful subsidies for the circular economy (HCE), half of them fall within the Energy class, and the remining ones roughly divided between VAT concessions (14 subsidies) and other subsidies (12), see Figure 1. None of the subsidies falling within the agriculture and fishing sector resulted in clear contrast with the principles of the CE. Regarding the financial value (Figure 2), all subsidies in contrast with the CE summed up to at least 13.5 billion in 2019, 58% concentrated in the Energy class with 7.8 billion euros, followed by 20% in the VAT concessions class (3.8 billion), 7% in Transport (1.2 billion) and 3% (0.7 billion) in Other subsidies class (3%).

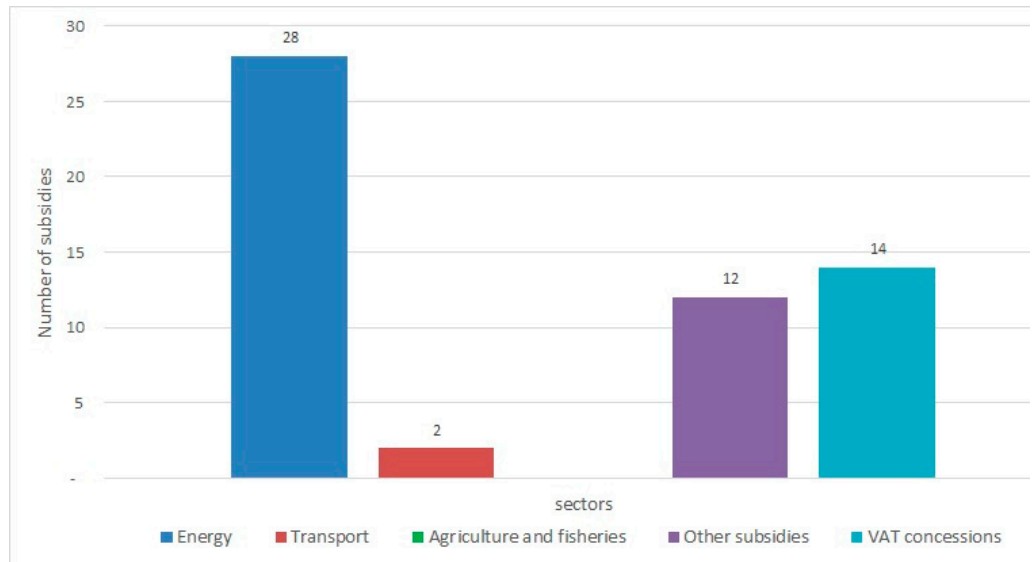

**Figure 1.** Subsidies that are harmful for circular economy (HCE), number of subsidies (2019).

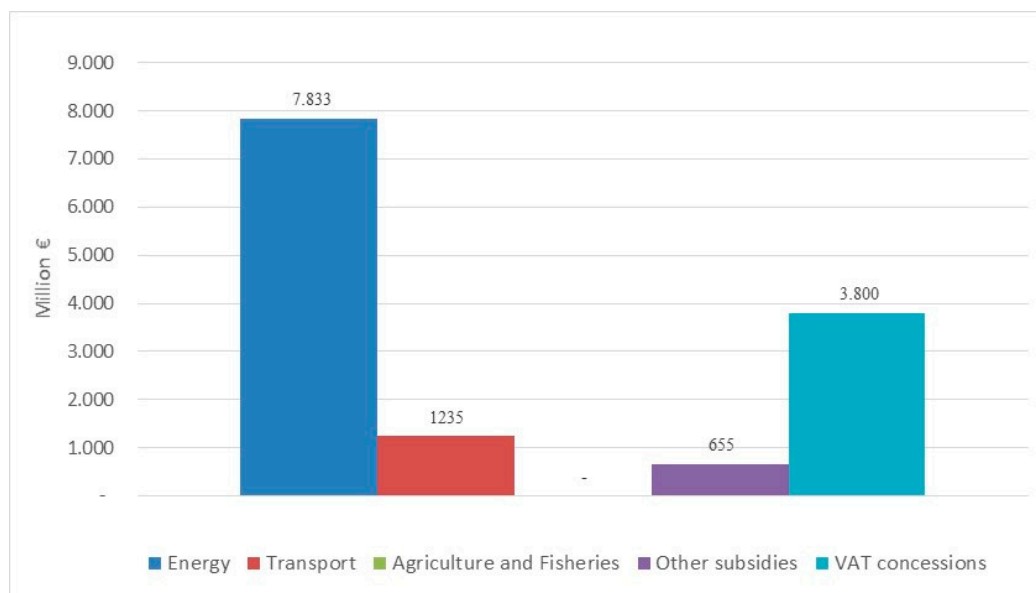

**Figure 2.** Subsidies that are harmful for circular economy (HCE), financial impact (million euro, 2019).

### 4.1.1. Energy

In this sector, the policy coherence analysis highlights a great number of subsidies in contrast with the CE (28 out of 50 subsidies examined in the energy sector). The most of them are indirect subsidies, mainly related to exemptions or concessions compared to ordinary rates of excise duty on energy products (see SM 2 Table S1). As expected, the most concerned CE principle is that of efficiency in the consumption of natural resources (fossil fuels subsidies): the subsidy lightens the energy price signal on producers or consumers, reducing the economic drive to efficiently use a scarce resource and to develop innovative solutions based on renewable sources.

Subsidies being found HCE are all subsidies that the Catalogue considers harmful to the environment (EHS), with the only exception of a subsidy (EN.SI.22—Diesel and LPG used for heating in geographically or climatically disadvantaged areas: mountain areas, Sardinia, smaller islands), for which the Catalogue reports two different opinions for the two fuels concerned: harmful for diesel, friendly for LPG. This case has been counted as two different subsidies, LPG being friendly for the environment but still harmful for CE (LPG is obtained from fossil resources). The positive motivation of the Catalogue is based on the fact that Directive 2014/94/EU [69] includes LPG among the "alternative fuels" to be promoted in the EU (this Directive is an example where policies may offer inner contradictions, favoring "lock in" in fossil-based investments) and on the lower harmful emissions of LPG as compared to other fuels used in such areas.

Finally, it should be noted that as many as five energy subsidies that are classified as EHS in the Catalogue present an uncertain profile from the CE point of view (as it is possible to identify both contrasting and favoring trends on CE principles): they have been therefore excluded from the construction of SM 2 Table S1. Among these there is a subsidy of great financial value (5.2 billion euro in 2019) that is worth a comment: EN.SI.24—Different tax treatment between petrol and diesel. The Catalogue considers this "implicit" subsidy an EHS in the light of a study [74] that estimated greater specific environmental external costs (related to both air pollution and greenhouse gases damages) of the diesel car fleet in Italy compared to petrol in the 1990–2015 period, with the argument that the lower excise duty of diesel fuel as compared to petrol has favored over the years the growing-up of the diesel car share in Italy to the detriment not only of the "less polluting" (lower external costs) petrol cars but also of LPG, methane and electricity vehicles that use "alternative" fuels pursuant to Directive 2014/94/EU [69]. In the present paper, this subsidy has been classified as "uncertain" from the point of view of the CE, since this subsidy mainly concerns fuel consumptions (principle of efficient use of scarce natural resources) but the direction of the final effect on cars' fuel consumptions is highly uncertain. In fact, observing the strong evolution of the Italian car market in the last year (strong diffusion of a variety of hybrid and battery electric vehicles, reaching an overall share of 20.5% of yearly registrations in 2020 from 6.6% in 2019 [75]), on the one hand we can see the strong commercial preference by consumers for gasoline hybrid models with low specific consumption, rather than for diesel hybrid models (the sales of which are limited to the large car segment, with high specific consumption) [75,76]: from this point of view, the lower excise duty of diesel fuel risks curbing the growing spread of hybrid vehicles with low specific energy consumption (petrol hybrid vehicles). However, the notorious energy efficiency of diesel internal combustion engines (ICE) seems to go well far beyond the comparison with petrol ones, as in some cases diesel ICE models equal or outperform also petrol hybrid models of the same class [77,78]: from this second point of view, the lower excise duty of diesel fuel is rewarding the energy efficiency of diesel ICE cars, but due to the increasing sales of low consumption petrol hybrid cars (first argument) there is uncertainty about the final effect on the fossil fuel consumptions of the car fleet.

### 4.1.2. Transport

Only 2 of the 16 subsidies examined in the transport sector have been classified as incompatible with the principles of the CE (SM 2 Table S2):

- TR.SI.04—Tax concessions on company car fringe benefits, since this subsidy favors the purchase of large displacement cars and their higher use by the employee (higher yearly mileages, as stated by the Catalogue);
- TR.SD.05—Scrapping fund for rail freight wagons, which in the Catalogue is considered an EFS (new rail wagons increase the quality of the transport service, thus favoring the use of the more sustainable rail transport as compared to road transport of goods); from the point of view of the CE the subsidy reduces the useful life of wagons.

In the transport sector the CE perspective leaves uncertainty in a quite high number of cases (5 uncertain subsidies, against only one in the Catalogue). Emblematic examples are two similar subsidies that provide respectively full exemption (TR.SI.09) or a discount (TR.SI.10) on the annual car ownership tax depending from the car age: from one side these models have poorer energy efficiency technologies as compared to new ones in similar market segments, while from the other side the subsidy favors life extension and use of old cars.

Many other subsidies in the transport sector have been considered coherent with the principles of the CE (e.g., car bonus, see Section 4.2.2).

### 4.1.3. Agriculture and Fishing

None of the 44 subsidies examined in this sector was found to be in clear contrast with the principles of the CE. In the Catalogue 8 are the agricultural subsidies classified as harmful for the environment, mainly due to their negative effects on biodiversity: 4 of them appear neutral in a CE perspective (NCE), while the other 4 are classified as friendly (FCE) since they promote practices related to the reuse of biodegradable waste (avoiding the use of chemical fertilizers) or they extend the duration of the crop system (fight against the bacterium *Xylella fastidiosa*). In general terms, many agricultural subsidies simply aim at maintaining existing agricultural systems, preventing land use change and urbanization processes and these features tends to lead to a friendly classification from the point of view of the CE (see Section 4.2.3).

### 4.1.4. Other Subsidies

In this sector (41 subsidies examined), as many as 12 measures have been identified in contrast with the CE principles (SM 2 Table S3).

A first group of HCE subsidies concerns water resource (AL.SI.05-06-07, AL.SD.02); for example, specific provisions of the water tariff that have important social purposes (tariff allowance on the first consumption bracket, water bonus for poor households), according to the Catalogue are environmentally harmful since they reduce the resource scarcity economic signal provided by the water price and induce to a less efficient use of water. A second group concerns tax exemptions or deductions in the real estate sector (AL.SI.01, AL.SI.03, AL.SI.12), that are in contrast with the CE as they favor the construction of new buildings rather than renovating or using the existing ones, thus increasing the consumption of land, which is a scarce natural resource par excellence [79,80]. The only harmful subsidy concerning the waste sector is AL.SI.04—Reduction of the ordinary levy on solid urban waste disposed of in incineration plants without energy recovery. According to Law 549/1995 incineration plants without energy recovery benefit of an 80% discount on the landfill disposal tax rate, that is fixed by the Regions. This is clearly a harmful subsidy for the CE, since incineration of waste without energy recovery is at the bottom of the waste hierarchy and the tax relief penalizes the competitiveness of the recovery of waste materials. It should be noted that the Italian law doesn't include incineration with energy recovery in the waste disposal tax; even if this exclusion is not part of the Catalogue definition of a subsidy [19], taxing also this form of incineration would represent an option for a tax reform inspired by the CE paradigm [32].

### 4.1.5. VAT Allowances

Of the 23 subsidies that fall into this category, 14 have been classified as HCE (see SM 2 Table S4). Six of them concern energy products (methane gas, LPG, raw mineral oils, petroleum products and electricity). They are considered HCE with the motivation of the reduction of the scarcity signal, which leads to a relatively less efficient use of the concerned energy products. Even if some of these VAT allowances have been introduced with a social inclusion purpose, they benefit all consumers, not just the poorest, favoring a generalized less efficient use of the energy products concerned. A specific comment is worth for the two VAT allowances on electricity consumptions (IVA.07 and IVA.08). The last available data on the Italian electricity mix, produced by Terna [81], say that gross electricity production from renewable sources was 39.0% of the 2019 national electricity production (114 TWh on 292 TWh, excluding the contribution of hydroelectric pumping). In this paper, VAT allowances on electricity consumptions have been classified as harmful for the CE since they increase more fossil fuel consumptions than renewables—given that the share of renewable in the electricity mix is currently minor than the fossil fuel share. According to the Italian Energy and Climate Plan—INECP [64], the needed trajectory of Italy's renewable energy share on gross domestic consumption at 2030 will exceed 50% between 2028–2029 (reaching 55% in 2030).

One VAT allowance is related to the waste sector and is very relevant for CE: it is the 10% VAT allowance (instead of 22%) for waste disposal in landfills, which undoubtedly constitutes an HCE subsidy, given that it favors a form of disposal that is at the bottom of the priority list of the waste cycle.

### 4.2. Subsidies That Are Friendly for the Circular Economy (FCE)

The analysis of policy coherence with CE made it possible to identify 75 friendly subsidies for the circular economy (FCE), of which 30 in the Agriculture sector, 23 in Other sectors, 9 in Energy, 7 in Transport and 6 in VAT allowances (Figure 3). In terms of financial value of the subsidies (Figure 4), FECs summed up to at least 13.0 billion, 42% of which concentrated in the Energy sector, with over 5.4 billion euros, followed by Other subsidies with 25% (3.3 billion), Agriculture and fishing with 21% (2.7 billion), VAT allowances with 11% (1.4 billion) and Transport with 1% (0.2 billion).

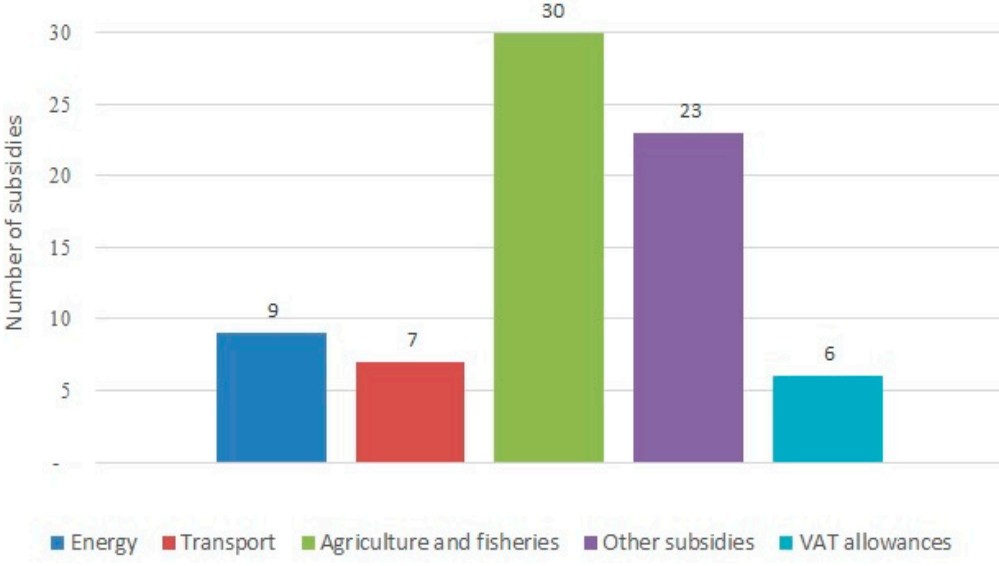

**Figure 3.** Subsidies that are friendly for circular economy (HCE), number of subsidies, 2018.

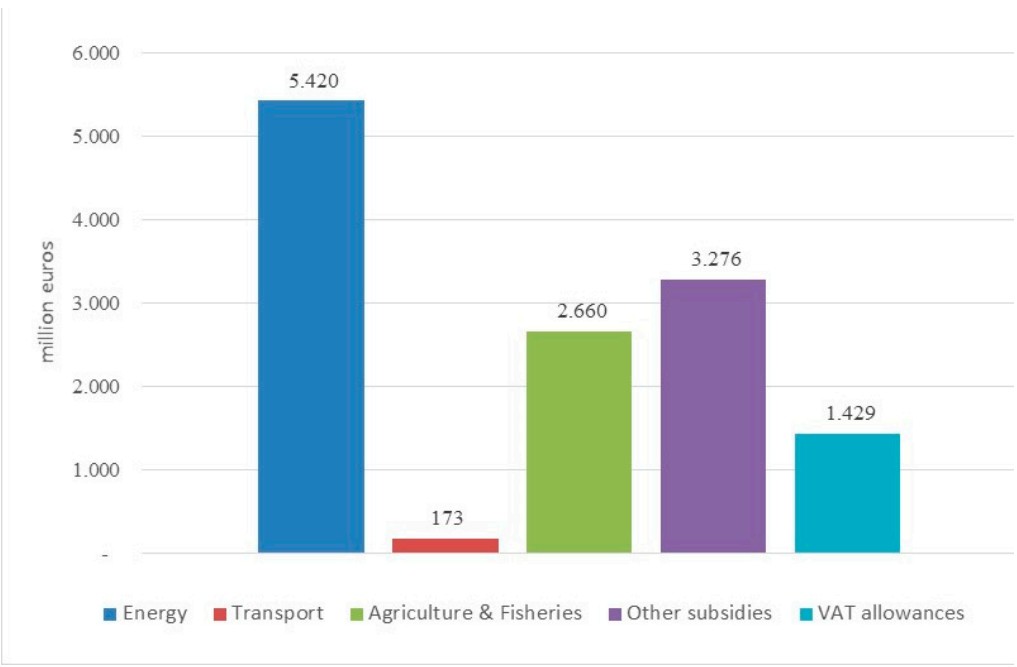

**Figure 4.** Subsidies that are friendly for circular economy (HCE), financial value (million euro, 2019).

### 4.2.1. Energy

The analysis of policy coherence of subsidies in the Energy sector led to the identification of 9 friendly subsidies (FCE) out of the 50 examined in the same sector (SM 2 Table S5). A high correspondence was found with the environmentally friendly subsidies (EFS) of the Catalogue: among the 9 FCEs, 8 are EFS, none are EUS (uncertain from an environmental point of view) and only one is classified EHS in the Catalogue.

Two FCE subsidies in this sector are of great importance, particularly the first for its financial value:

- EN.SD.06—Incentives for electricity produced from renewable sources other than photovoltaic, Ministerial Decree 23 June 2016 (5052 million euros in 2019 [82]) (this is not the last incentive scheme for renewable sources, the last edition of the Catalogue [19] covers subsidies in force until the end of 2018);
- EN.SD.08—Promotion of energy efficiency and energy production from thermal RES–Thermal account 2.0, Inter-ministerial Decree of 16 February 2016 (264 million euros in 2019 [83]).

The first mechanism covers a wide range of renewable energy sources. It has been classified FCE for consistency with the following CE principles (they differ according to the renewable energy source type that is subsidized by the M. D. of 23 June 2016) all others being considered not harmful:

- for thermodynamic solar, wind, ocean, hydroelectric, geothermal energy: use of renewable sources and saving of fossil fuels (in the case of hydroelectric plants, the art. 4 par. 3 of the Decree introduces many water protection restrictions).
- for biogas from anaerobic digestion of waste and sludge: use of renewable sources, saving of fossil fuels and compliance with EU waste hierarchy (the production of biogas is part of a process that gives priority to sludge recycling as compost for crops fertilization);
- for bioliquids obtained from agricultural residues: use of renewable sources to replace fossil fuels and saving of material natural resources for producing fuels. Bioliquids must currently comply with the Sustainability criteria required by Fuel Quality Directive 2009/30/EC [84] transposed in Italy through art. 38–39 of Leg. Decree 28/2011. The Directive 2018/2001/EU on the promotion of renewable sources (RED

II [68]) which has yet to be transposed in Italy (deadline in June 2021), favors second generation biofuels obtained from waste streams and has further strengthened the sustainability criteria for biofuels, bioliquids and biomass fuels;

- for biomass obtained from by-products (from the processing of forest products, forest management, public green pruning, straw and agricultural residues, animal by-products not intended for human consumption, agro-industry and wood processing by-products): use of renewable sources to replace fossil fuels and saving of material natural resources for producing fuels.

The second mechanism (EN.SD.08—Thermal account 2.0) encourages interventions to increase the energy efficiency of buildings and small plants for the production of thermal energy from renewable sources (solar thermal, heat pumps, biomass plants). The beneficiaries are mainly public administrations, but also businesses and households. The Thermal account appears coherent with the principles of the CE, favoring the use of renewable energy sources and/or a more efficient use of fossil fuels. When the incentive mechanism provides for the replacement of existing heating systems in private buildings, the regulatory framework [85] requires a certificate to guarantee their correct recovery and waste management. As for biomass plants, the same regulation favors the use of certified pellets or, as an alternative, the use of by-products of wood processing, avoiding the use of biomass directly produced from dedicated crops or forestry.

Regarding the only FCE subsidy that is harmful from the environmental point of view (EN.SD.02—Aid to operators at risk of carbon leakage), the change of the sign of the classification is due to the fact that the subsidy is not provided to operators without any conditionality, as assumed by the Catalogue. Art. 10 of Law 221/2015 regulating the subsidy adopts the ISO 50001 certification (energy management system) as the main criterion for awarding the subsidy to the operator, the purpose of which is to enable the organization to follow a systematic approach in achieving continuous improvement of energy performance and of the energy management system [86]. This meets the criterion of the CE of an efficient use of energy resources.

A specific mention deserves also the subsidy EN.SD.04b—Admission to the mechanism of Energy Efficiency Certificates of Organic Rankine Cycle systems for the self-production of electricity, as it is directly aimed at exploiting the waste heat of energy-intensive industrial processes: waste heat is a valuable by-product of certain industrial process, thus heat recovery is an important application of the CE, that allows the energy saving of natural scarce resources (fossil fuels).

Lastly, due to its financial importance, a specific mention deserves EN.SD.07—The energy account for photovoltaic system" (5970 million euros in 2019 [82]). This mechanism has been classified as uncertain for circular economy (UCE) due to contrasting effects on the CE:

- from the positive side, the mechanism promotes the use of renewable sources and saving of fossil fuels. In addition, a national law has prohibited since 2012 to subsidize ground-mounted photovoltaic systems in agricultural areas (rather than in already urbanized areas or on the roofs of buildings), preventing land consumption. Moreover, the mechanism has been conceived to manage the end-of-life phase of the photovoltaic (PV) panels (WEEE—Waste of Electrical and Electronic Equipment). In fact, national legislation has introduced since 2011 the responsibility of producers on PV panels at the end-of-life (M.D. of 5 May 2011, M.D. of 5 July 2012, Leg.D. 49/2014), to guarantee the financing of an adequate management of the collection, transport, treatment, recovery and disposal operations of waste deriving from photovoltaic panels, also through the use of a collective system (recycling consortia);
- as regards the negative aspects, the mechanism appears to be in contrast with the principle of efficiency in resource use. In fact, there is evidence in the scientific literature regarding the high consumption of scarce natural resources in the production cycle of cells and PV modules. For example, a recent LCA study [87] has shown that, for the same amount of energy produced, the 2017 national mix of PV plants causes

a higher consumption of abiotic resources than gas, coal or oil based thermal plants. This result is mainly due to the production processes of zinc concentrate and the silver extraction processes. Zinc is used for galvanizing in the production of aluminum supports for solar panels and of aluminum alloys used in the panel. The method used to calculate the "mineral, fossil and renewable resource" impact indicator is the one developed by CML Leiden [88], according to which the scarcity of each substance extracted from the natural system is calculated as the annual extraction of the substance divided by its availability squared.

### 4.2.2. Transport

As regards the transport sector, seven subsidies have been classified as coherent with the CE, mainly for the lower primary energy consumptions allowed by vehicles benefiting from the incentives, see SM 2 Table S6.

In particular, the positive classification of the following subsidies, concerning the promotion of electric mobility, deserve an in-depth comment:

- TR.SI.07—Reduction of the car property tax for electric vehicles;
- TR.SI.11—Tax deductions for the purchase and installation of charging infrastructure for electricity powered vehicles;
- TR.SD.06—Contribution for the purchase of a brand new electric or hybrid two-wheeled vehicle (two wheels bonus);
- TR.SD.07—Contribution for the purchase of new low $CO_2$ emissions car (car bonus).

One of the main objections to subsidies for electric vehicles (EV) from a CE perspective is related to the proper management and recycling of end-of-life batteries. As compared to conventional lead-acid batteries, current lithium-based electric car battery packages are much heavier [89] and more difficult to manage and treat, mainly due to the flammability and toxicity of lithium. Other metals used in lithium-based batteries, such as cobalt, nickel, and manganese are considered toxic heavy metals and raise concerns in waste management [90]. A 2018 report by Ricardo for the European Commission [91], the results of which were subjected to consultation with main stakeholders, analyzed the potential environmental impacts and hotspots of EV batteries over their whole life cycle if deployed at large scale in the EU. The report also evaluates the strengths and weaknesses of the EU industry and waste policy for dealing the transition to electric mobility. According to this report, theoretically most materials in a lithium-ion battery can be recycled. Recycling rates with current pyrometallurgical processes (designed for lead-acid batteries) are around 50–60% of total weight (with 50% required by the Batteries Directive 2006/66/EC); higher recovery rates of 70–80% are considered feasible in the medium term by using new purely hydrometallurgical processes. The report highlighted an increasing industrial interest for reuse (for the same type of electric vehicle) or repurposing (second-life use for energy storing) of the EV batteries prior to the recycling stage, with a significant potential for a reduction of their life cycle emissions and consumptions of primary raw materials. Ricardo report sees favorable CE perspectives also for the strength of European industry in battery recovery and recycling, driven by strong EU legislative requirements, such as the Batteries Directive, the End-of-Life Vehicle Directive (2000/53/EC) and REACH Regulation (EC 1907/2006). Italian recycling industry is also in a good position to successfully manage the new challenge of end-of-life EV batteries; for example, according to Eurostat's last available data [92], Italy is one of the leading Member States for recycling efficiency for lead-acid batteries in 2018. Overall, we can say that perspectives for increasing the re-use and recycling rates of EV batteries and assuring their safe management in Italy (and in the EU context) are quite positive.

Another objection to subsidies for EVs under a CE perspective is related to possible indirect effects on primary resource consumptions. As regards fossil primary resources, the Catalogue itself acknowledges that the LCA carbon footprint of EV (therefore considering the fossil fuel consumptions needed for the production of both the vehicle and the energy products used during the vehicle's lifetime) in the EU is on average 55% lower than

conventional cars [19] (p. 295). As regards all primary mineral and energy sources, a recent LCA report [93] highlights that the value of the indicator "Depletion of mineral, fossil and renewable resources" calculated in the case of the battery electric model is about 50% lower than that of petrol, diesel and methane car versions, while in the case of the plug-in hybrid model this advantage reduces to 15%. The report uses the same indicator of resource depletion previously mentioned, based on CML method [88], that weighs according to a resource scarcity algorithm the resource flows taken from the environment in the different processes of the life cycle (vehicle production, battery production, production of fuels, their combustion during the use of the vehicle, the production of spare parts for the maintenance of the vehicle, the production and maintenance of the infrastructure used by the vehicles).

For the two subsidies based on the simultaneous scrapping of an old car (TR.SD.07) or of a motorbike (TR.SD.06), the analysis of coherence with the principles of CE is made more complex by the anticipated end-of-life of vehicles, that goes in the opposite direction of CE. The reduction of the expected mileage of the currently owned car, due to the subsidy, generates a loss of resource efficiency along its life cycle (in tends to increase the LCA resource consumption indicator per km traveled by the vehicle). However, according to the comparative results obtained for electric vehicles by [93] for the indicator "Depletion of mineral, fossil and renewable resources" it can be said that the net effect would probably remain positive (the saving of resources offered by new plug-in vehicles, would not be compensated by the "waste" of resources due to the anticipated scrapping of existing vehicles). As regards two-wheeled vehicles (TR.SD.06), the bonus mechanism with scrapping could lead to similar net benefits, as highlighted by LCA research on electric mopeds [94]. We can conclude for a positive classification of the above-mentioned subsidies under a circular economy perspective (FCE).

### 4.2.3. Agriculture and Fishing

As already mentioned in Section 4.1.3, while the present analysis did not find HCE, it has identified 30 FCE on 44 agricultural subsidies examined (see SM 2 Table S7). In general terms, this is mainly due to the recurrent aim of agricultural subsidies to prevent the abandonment of land and to favor the protection of valuable ecosystems in agricultural areas, with a positive effect in preventing the urbanization phenomena and land consumption.

It is possible to distinguish the following groups of agricultural subsidies in favor of the CE:

- The group of EHSs in support of animal husbandry in its various types (AP.SD.03—Beef cattle, AP.SD.09—Dairy cattle and AP.SD.10—Dairy buffalo), where the subsidy also indirectly encourages production of animal slurry which, according to current Italian legislation, can be excluded from the legal regime of waste if it is subject to "agronomic use". This legislative provision promotes the reuse of slurry from intensive farming to fertilize the fields, avoiding the use of chemical fertilizers. This agricultural practice has deep historical roots (albeit with different volumes at stake from the current ones) and is consistent with the recycling principle of the circular economy (FCE); but it must be underlined that this is a case of contrasting outcome when applying the environmental perspective, since the Catalogue [19] (pp. 149–150) classifies this subsidy as harmful for the environment (EHS), due to the significant ammonia emissions of such practice, with harmful effects on health.
- A large group of agricultural subsidies that the Catalogue considers uncertain from an environmental point of view (EUS), but which from the point of view of the CE play an important role in preventing the abandonment and urbanization of marginal land (for example, this group includes AP.SD.11—Support for dairy livestock in mountain areas, AP.SD.17—Support of agricultural practices beneficial for the climate and the environment, AP.SD.20—Specific support for beef cattle breeding (suckler cows) and AP.SD.21—Specific support for sheep and goat livestock).
- Another group is made of agricultural subsidies, impacting on biodiversity (EHS in the Catalogue), that under a CE perspective have the positive function of protecting

and renovating traditional farming systems and related economic activities, such as olive groves (AP.SD.08, AP.SD.19 and AP.SD.32-33) or vineyards and wine production (AP.SD.34-36). An emblematic example is constituted by AP.SD.08—Measures for the relaunch of the olive sector in areas affected by Xylella fastidiosa, which from the point of view of the CE extends the useful life of the crop system and prevents abandonment of agriculture in an area (Puglia region) that is subject to high rates of land consumption [79]. The Catalogue classifies this subsidy in the opposite way (EHS) by stating that "the measure encourages replanting with a plant type that is tolerant to the bacterium, a practice that favors a reduction of species diversity by exposing them to new epidemics in the future. The goal should be to diversify in genetic terms to minimize future risk" [19] (p. 148).

Lastly, it is interesting to mention AP.SD.39—Support for beekeeping which, from the environmental point of view of the Catalogue is an EFS (it allows the protection of natural pollination, an essential ecosystem service that contributes to our wellbeing and to fertilization of certain crop types) and from the point of view of the CE is favorable as well (FCE), since it avoids the consumption of abiotic resources needed by artificial pollination methods [95]. The "nature-based solutions", allowing in this case the sustainable integration of modern agriculture and nature [96], are excellent solutions also for the purpose of the circularity of the economy.

### 4.2.4. Other Subsidies

In the "Other subsidies" sector, the Catalogue includes a group of waste management subsidies which are particularly relevant for the CE (SM 2 Table S8):

- AL.SD.05—Increase from 20% to 100% of the revenue recycling percentage required by the law establishing the regional tax on waste disposed in landfills and incineration plants (TARI): the revenues of the tax are allocated to interventions for the CE [18] (art. 34);
- AL.SI.25—Provisions to promote prevention policies in the production of waste: Municipalities are enabled to introduce discounts on TARI to non-domestic users who achieve waste production savings as the result of preventing actions and recycling of the organic fraction [18] (art. 36);
- AL.SI.28—Waste tax allowances to avoid food waste: allowances are provided to non-domestic users who donate food to poor people (art. 17 Law of 19 August 2016, n. 166);
- AL.SI.20—Green garden bonus: a 36% deduction from the personal income tax of the cost for small greening actions in private uncovered areas of existing buildings (art. 1 c. 12-15 Law of 27 December 2017, n. 205, that is the budget law for 2018);
- AL.SI.29—Tax credit for purchases of mixed plastics (plasmix) from separate collection: company income tax credit of 36% for the purchases of products made with mixed recycled plastics aimed at promoting the recycling of plastics (art.1 c. 96-99 of budget law for 2018);
- AL.SI.33—Company income tax credit of 36% for the purchase of products made with recycled plastic, biodegradable and compostable packaging or packaging made with recycled paper or recycled aluminum (art. 1 c-73-77 Law of 30 December 2018, n. 145, that is the budget law 2019);
- AL.SI.31 and AL.SI.32: two company income tax credits, respectively on the recovery and reuse of packaging by the selling company and on the purchase of compost or intermediate products with a recycled content of 75% at least, introduced by art. 26bis and 26ter Law of 28 June 2019 n.58.

From the above list of subsidies FCE it seems that a new "policy season" of support measures directly inspired by the CE emerges since Law 221/2015 [10] and continues in the following years through specific interventions scattered in different "trains of legislation" (e.g., "budget laws"). In a national context of high taxes and tax evasion such as the Italian one [66], a common feature of most of these new measures is the use of tax discounts

instead of direct subsidies to incentivize company level CE practices, and precisely in the key decision-making areas of procurement of materials and product packaging (AL.SI.29, 32, 33), sales and deliveries (AL.SI.31) and waste management (AL.SI.25, 28). Given the European Green Deal commitments to strengthen legislation on the CE [35,57], the elaboration of a national plan for CE and of a more dedicated legislative framework supporting the implementation and monitoring of such mechanisms would be helpful.

4.2.5. VAT Allowances

Six subsidies friendly for the CE (FCE) out of the 23 subsidies falling in this category have been identified. The main ones belong to a group of three VAT allowances (10% instead of 22%) concerning building renovations (SM 2 Table S9):

- IVA.20—Building repair and renovation services of private homes;
- IVA.14b—Provision of services dependent on procurement contracts relating to building restoration interventions;
- IVA.19—Leases of renovated residential buildings carried out by the construction companies.

These subsidies have a common profile coherent with many CE principles, given that they are aimed at repairing, renovating and restructuring existing buildings (thus extending the building lifetime and avoiding new land consumption) or at reducing the non-use periods of existing buildings which, from the point of view of resources needed to provide the housing service, constitute an environmental inefficiency.

*4.3. Comparison of the Results with the Environmental Subsidies of the Catalogue*

In this chapter a comparison is made between the environmental assessment of the Catalogue and the circularity perspective applied in the present analysis, in order to provide a more systematic information on the different outcomes obtained when the two perspectives are separately applied for the evaluation of subsidies. The results are illustrated for subsequent steps of analysis, starting with those subsidies successfully classified as harmful or friendly under the two perspectives. As anticipated in Section 3.1, the starting point is the same: the set of subsidies examined by the third edition of the Catalogue, updated to year 2018, that is made of 174 subsidies. The reference year for the comparison is thus 2018.

Referring to Figure 5, while the Catalogue has successfully classified (as EHS or EFS) the 85% of the considered subsidies, the analysis of policy coherence has led to the classification in HCE or FCE in 75% of cases. If we consider the financial weight of the subsidy, while the Catalogue has reached a classification in EHS or EFS for 80% in value (35.1 billion euros), the circularity classification in HCE or FCE corresponds to 59% in value (25.9 billion). However, the aggregated financial results should be considered with caution due to uncomplete filling of the financial field of the Catalogue's subsidy identity cards. Just to provide a comparison of the order of magnitude of average financial values obtained, the 2018 average of the whole set of harmful or friendly subsidies identified with the circularity perspective is 198 million euros per subsidy, that is 16% lower than that of the Catalogue (237 million euros).

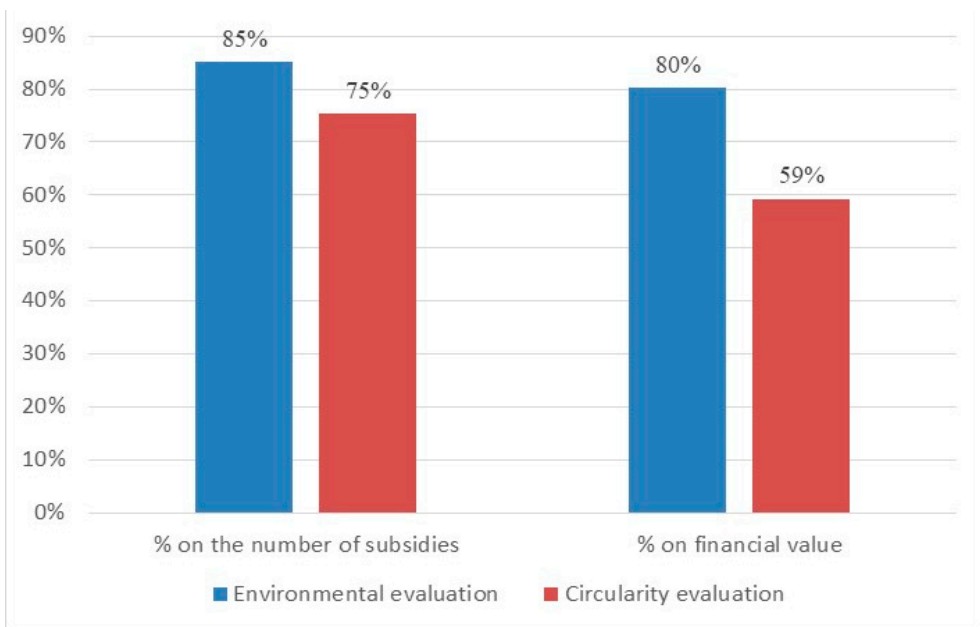

**Figure 5.** Harmful or Friendly subsidies: comparison between the environmental evaluation of the Catalogue and the circularity evaluation (% on the number of subsides and % of the financial value).

The discriminating ability of the CE perspective in the evaluation of subsidies (ability to identify harmful subsidies, worthy of reform, or friendly subsidies, worthy of confirmation) is comparatively lower than the environmental perspective one, managing to discriminate (successfully classify) a relatively lower percentage of subsidies. This is confirmed also by the number of "uncertain" subsidies, which is higher in the CE analysis than in the environmental evaluation of the Catalogue (respectively 31 against 26 uncertain subsidies). The higher uncertainty related to the CE perspective is often due to the inner contradiction between different CE principles (for example between the extension of products' lifetime and innovation to increase energy efficiency; or between energy efficiency and land and materials consumption in subsidies for buildings renovations with increased volumes).

As regards the comparison between sectors, in the Energy and Transport sectors the circularity perspective allows to successfully classify (harmful or friendly subsidies) a lower number of cases than the environmental perspective, while in the other sectors the two perspectives seem to be equivalent for their discriminating ability (Figure 6).

4.3.1. Harmful Subsidies Comparison

If we consider only the subsidies classified as harmful, the environmental evaluation of the Catalogue identified 72 cases, for a value of at least 19.7 billion euros in 2018, while the circularity evaluation identified far fewer cases (56), for at least 13.2 billion euros in 2018 (Figure 7). The average yearly value of harmful subsidies identified with the circularity perspective is 235 million euros per subsidy, that is 14% lower than that of the Catalogue (273 million euros). The lower discriminatory ability of circularity in identifying harmful subsidies is confirmed, in comparison with the environmental perspective.

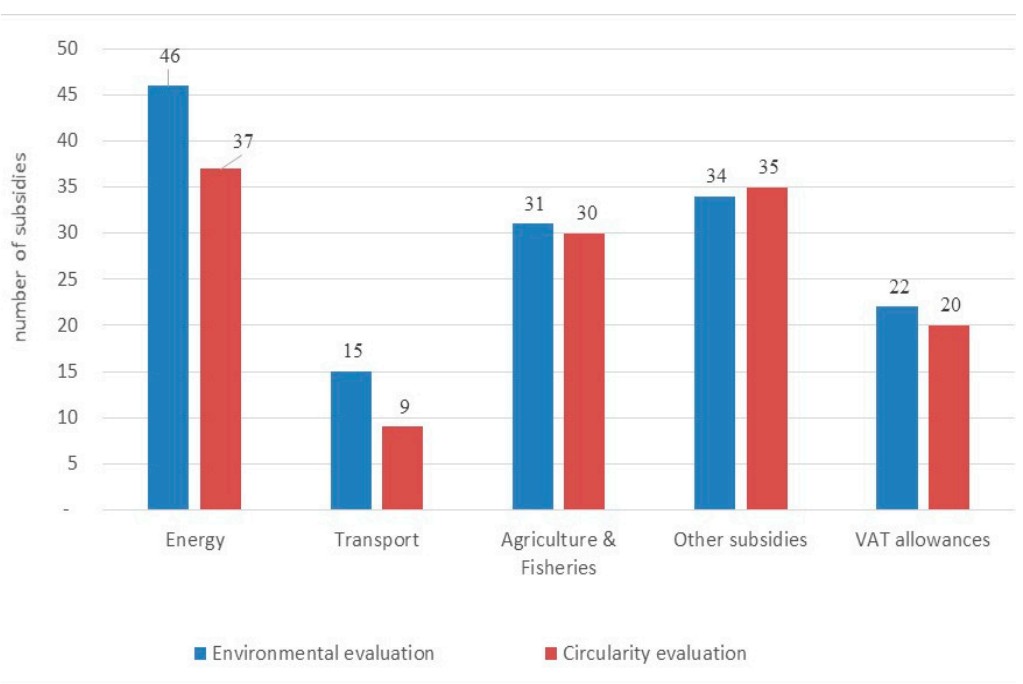

**Figure 6.** Harmful or Friendly subsidies by sector: comparison between the environmental evaluation of the Catalogue and the circularity evaluation (number of subsides).

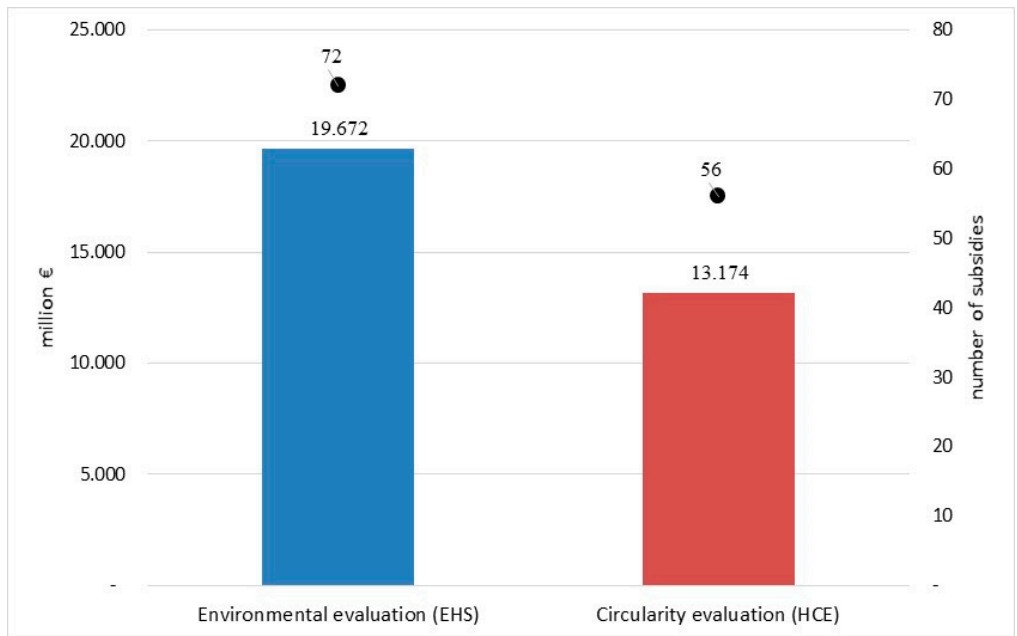

**Figure 7.** Harmful subsidies: comparison between the environmental evaluation of the Catalogue and the circularity evaluation (number of subsides and total financial value).

In the sectoral breakdown of the number of subsidies (Figure 8) the greatest decrease in harmful subsidies by changing perspective from the environmental one to circularity occurs in the agriculture sector (-8 cases), followed by Energy (-5), VAT allowances (-3) and Transport (-2), while in Other subsidies there are 2 additional cases of harmful subsidies. As said in Section 4.1.4, the "Other subsidies" class includes harmful subsidies mostly related to the real estate construction sector (adverse effects on land and resource consumptions) and water consumption (less efficient use of water resource).

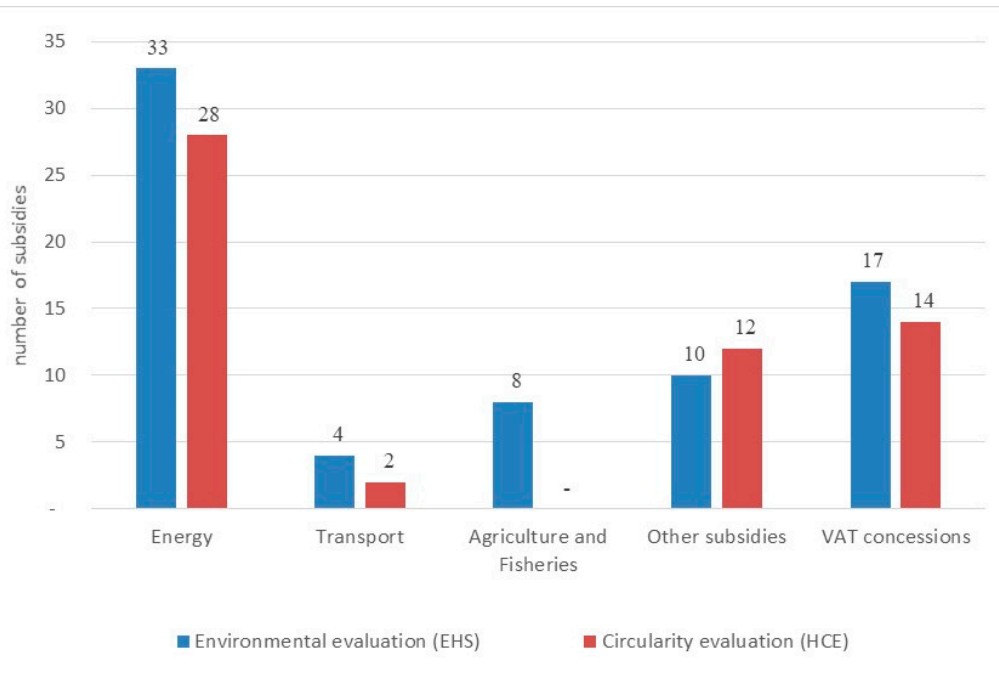

**Figure 8.** Harmful subsidies by sector: comparison between the environmental evaluation of the Catalogue and the circularity evaluation (number of subsides).

### 4.3.2. Friendly Subsidies Comparison

Moving on to friendly subsidies, the environmental evaluation of the Catalogue identified 76 cases, for an aggregated value of at least 15.4 billion euro in 2018, while the circularity evaluation identified a similar number of cases (75), for a lower total financial value, equal to at least 12.7 billion euros in 2018 (Figure 9). The average yearly value of friendly subsidies identified with the circularity perspective is 170 million euros per subsidy, that is 16% lower than that of the Catalogue (202 million euros).

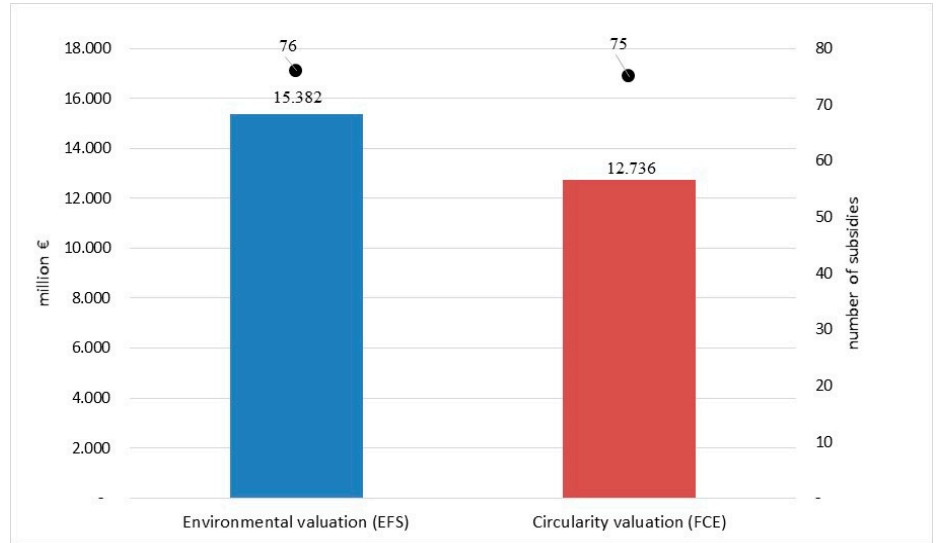

**Figure 9.** Friendly subsidies: comparison between the environmental evaluation of the Catalogue and the circularity evaluation (number of subsides and aggregated financial value).

The ability to discriminate friendly subsidies of the two valuation perspectives appears to be equivalent, with a slight aptitude of the environmental perspective to capture subsidies with relatively greater financial importance. In the sectoral breakdown (Figure 10),

great differences in the comparison between the two perspectives stand out in three sectors: the move from the environmental perspective to that of the CE leads to an increase in the number of friendly subsidies in Agriculture (+7 subsidies), while in the two sectors of Transport and Energy there are 4 fewer cases of friendly subsidies. The circularity perspective seems to reward subsidies in the agriculture sector with a more positive classification, while it is less generous in the transport and energy sectors.

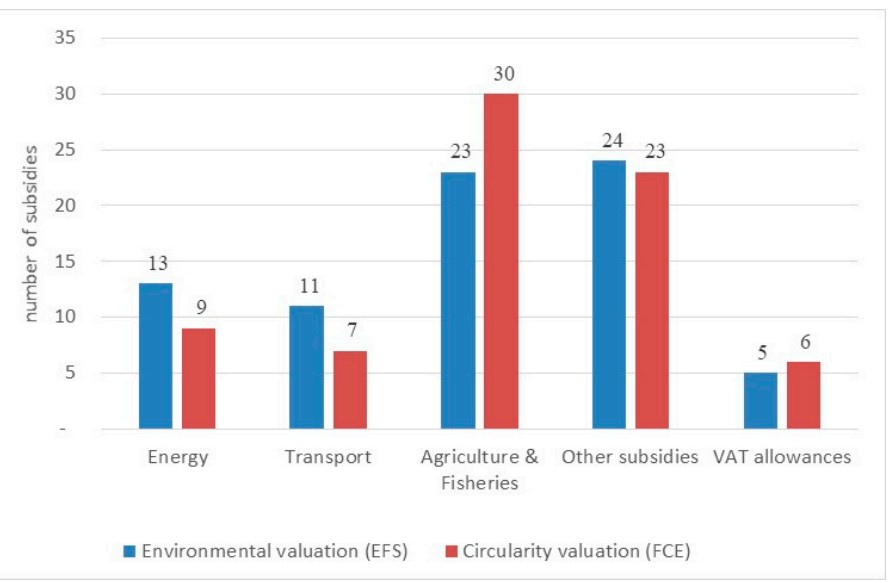

**Figure 10.** Friendly subsidies by sector: comparison between the environmental evaluation of the Catalogue and the circularity evaluation (number of subsides).

### 4.3.3. Detailed Comparison between the Environmental Assessment and the Circularity Assessment

Table 1 shows the detailed comparison of the outcome of Catalogue's environmental evaluation with that of circularity analysis of the present work, highlighting all the typological occurrences in terms of number of subsidies.

**Table 1.** Detailed comparison between the environmental assessment and the circularity assessment, by outcome combinations.

| Comparison with [19] | Number of Subsidies | % |
|---|---|---|
| EHS-HCE | 51 | 29.3% |
| EUS-HCE | 3 | 1.7% |
| EFS-HCE | 2 | 1.1% |
| EFS-FCE | 61 | 35.1% |
| EUS-FCE | 8 | 4.6% |
| EHS-FCE | 6 | 3.4% |
| EHS-UCE | 10 | 5.7% |
| EUS-UCE | 14 | 8.0% |
| EFS-UCE | 7 | 4.0% |
| EHS-NCE | 4 | 2.3% |
| EFS-NCE | 3 | 1.7% |
| EUS-NCE | 0 | - |
| Subsidies not anymore in force in 2018 | 5 | 2.9% |
| Total | 174 | 100.0% |

Legend: EHS: Environmentally Harmful Subsidy; EFS: Environmentally Friendly Subsidy; EUS: Environmentally Uncertain Subsidy; HCE: Subsidies that are Harmful for the Circular Economy; FCE: Subsidies that are Friendly for the Circular Economy; UCE: Subsidies that are Uncertain for the Circular Economy; NCE: Subsidies that are Neutral for the Circular Economy.

Subsidies for which the positive environmental evaluation has been confirmed by the analysis of policy coherence with the CE (EFS-FCE) are the most frequent case, with 35.1% (61 subsidies), followed by subsidies for which the negative environmental assessment has been confirmed from the point of view of the CE (EHS-HCE), with 29.3% (51 cases).

By re-aggregating the detailed results of Table 1 into broader classes (see Figure 11), all subsidies that have had a confirmation of the outcome of the assessment (EHS-HCE, EFS-FCE, EUS-UCE) are 126 (72.4%), while those that have undergone a change in the classification (the remaining ones, except for the 5 subsidies no longer in force) are 43 (24.7% of the total). In particular, a complete "reversal of judgment" (from EHS to FCE and from EFS to HCE) occurs in 8 cases (4.6%).

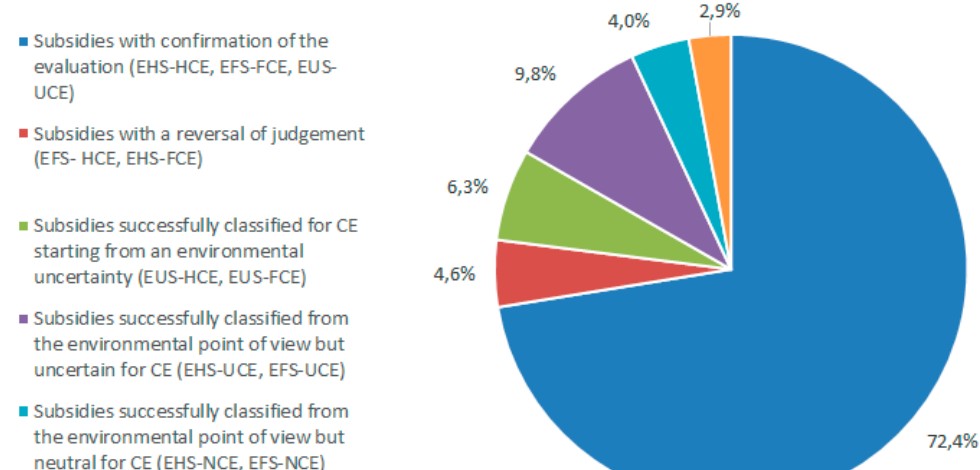

**Figure 11.** Re-aggregated comparison between the environmental and circularity evaluation, by category of variation.

The results of the comparison confirm that, despite the overlapping areas between the environmental and circular policies, the two paradigms also present autonomous areas of intervention that can place subsidy effects in partial or full contradiction with each other. Single cases of contradictory outcome between the Catalogue classification and this paper circularity evaluation can be easily found in the SM 2 tables on HCE and FCE, by comparing the information of the column related to the Catalogue's classification (EHS, EFS or EUS) with the information of the column related to CE qualification ("Circular economy principles harmed by the subsidy"—Tables on HCE; "Circular economy principles favored by the subsidy"—Tables on FCE).

Identifying subsidies that present trade-offs between the CE and a conventional environmental perspective, as achieved through the analysis of policy coherence made in this paper, can provide to policy-makers useful information to increase the consistency of existing subsidies with both the environmental and CE objectives, thus improving the effectiveness of sustainable development policies [2,17], which see the two paradigms "side by side".

## 5. Methodological and Analysis Limitations

The main limitation of the analysis carried out concerns the method used for evaluating subsidies: even if supported by scientific evidence (reports, LCA studies, indicators...), the analysis of policy coherence with the principles of CE remains a subjective approach, since as far as the author is aware suitable indicators are currently not available to measure homogeneously the impacts of subsidy schemes on the circularity of economic activities [8,97]. Proper indicators need proper accounts. In the case of subsidies the needed accounts should be developed at the firm and household/user level, that is the level of subsidy beneficiaries and where decisions on the use of resources are taken: while statistics

on waste recovery and recycling are diffused in developed countries, problems are faced at a global level, due to cultural differences and difficulties in reaching a global consensus on what is waste [98]; information on reuse, repair and refurbishing is lacking [8]; data for suggested CE indicators at the product design phase are totally missing [97]. Enhancing the company disclosure of resource-related data in non-financial reporting and increasing the role of resource labelling along the value chain (materials, components, products, waste recovery) [35] are essential drivers for the development of CE indicators at the company level. Indicators supporting the assessment of subsidies (effectiveness of subsidies on circularity or their harm to circularity) is an important area of further research.

Second, the "analysis of policy coherence" is a simple method that can be easily replicated by administrations; it does not pretend to substitute a full assessment of subsidies from an economic welfare point of view. If time and resources are available, particularly for subsidies of large financial size, it is recommended to integrate the suggested method with dedicated in-depth economic studies, possibly performing a full cost–benefit analysis (CBA) at the subsidy level rather than cost-effectiveness analysis [99]. The latter has serious limitations when applied to subsidies, since it is designed to calculate the intentional effect of the subsidy [99,100], without considering the unintentional effects, which can often be harmful for the environment or for the CE. Moreover, the concept of costs in cost-effectiveness analysis is often limited to the direct financial cost for the State, failing in considering all other social cost caused by the subsidy. Current practices with CBA at the policy level of subsidy schemes seem very engaging [101]. According to an OECD review of national administrations' practices [102], policy level CBAs are less diffused as compared to project level CBAs and with a narrower coverage of non-climate environmental impacts, suggesting that the risk of not being able to analyze all the main economic, social and environmental effects of a subsidy from a monetary point of view with a single CBA study is high.

A third remark is related to the boundaries of the policy coherence approach, that relies on official definitions and on subsidies' policy defined aims. The Taxonomy definitions of CE and of other environmentally sustainable objectives may not be optimal from a sustainability academic perspective such as, for example, the one based on the "planetary boundaries" [103,104]. Even if it's beyond the scope of the paper to make a full analysis, the Taxonomy definition of CE uses the terms "reducing the environmental impact" and doesn't include any "carrying capacity" concept; and one of the CE principles is to efficiently use natural resources, including sustainably sourced bio-based and other raw materials. Korhonen et al. [98] who attempted a critical analysis of the CE concepts with a planetary boundary approach, suggest an academic definition of CE that includes the following two key concepts: "Circular economy limits the throughput flow to a level that nature tolerates and utilizes ecosystem cycles in economic cycles by respecting their natural reproduction rates" [98] (p. 39). The use of bio-based natural resources and of organic waste should be increased (within the carrying capacity of eco-system services) rather than only "efficiently used". A bio-based CE is possible, but we need to increase research on the measurement of carrying capacity, consensus on standards and certification schemes [105,106]. Moreover, it must be reminded that the EU Taxonomy definitions are limited by its administrative applicability. Sustainable development needs a world-wide approach to CE, as well as finance.

Other limits of the analysis are related to the data set used. Over 170 subsidy schemes seem many, but a Governmental report produced in 2012 with the aim to cut tax expenditures in Italy [107] identified 720 measures in force in 2011. Even if the Catalogue [19] considered also this source in the screening of subsidies to be assessed, it applied a conventional environmental protection perspective and it is probable that some of the subsidies excluded from the Catalogue be relevant for the CE (potentially harmful of friendly for the CE). Moreover, the financial value of many subsidies assessed by the Catalogue is missing and this is an obstacle for a proper analysis. Lastly, the small number of direct subsidies as compared to indirect ones highlights an underestimation of direct subsidies. A

comprehensive coverage of direct subsidies and a completion of all fields of the subsidy identity card would increase the utility of the Catalogue for further research applications and policy decision making.

Many of these shortcomings can be overcome through an increased cooperation between national Ministries responsible for the sectors benefiting from subsidies. In Italy, the recent establishment of the Ministry of Ecological Transition through the unification of the Ministry of Environment with a section of the Ministry of Economic Development [108], is an opportunity to assure the needed coordination between CE and environmental protection policies. The commitment of the EU Green Deal for greening national budgets [57] can become a driver for Member States to enhance internal cooperation between administrations to find out green subsidies as well as harmful subsidies from both the environmental and CE perspectives.

The role of academic research and of cooperation with the national administrations in assessing subsidy schemes, starting from the new ones dedicated to CE targets, is of utmost importance. As mentioned in the discussion, the research agenda related to subsidy analysis is rich of research streams to be developed, ranging from CE accounting (particularly at the company level), to modelling of circularity, development of indicators that can be used in analysis of effectiveness of economic instruments, labelling of the materials' content in products, enhancing methods for assessing policy coherence and integration with other assessment tools, sustainability science applications to improve specification of policy objectives and definitions, standards for CE practices.

## 6. Conclusions

In this paper an analysis of policy coherence with the EU CE principles of a large set of environmentally significant subsidies (potentially harmful or friendly for the environment) currently in force in Italian legislation, has been carried out. The adopted approach is "freely inspired" by the OECD [16] and IEEP [22] methodologies developed for reforming environmentally harmful subsidies and by the 2030 Agenda [2] recommended method of Policy Coherence for Sustainable Development, originally elaborated by the OECD [17] to help Government in "aligning" sectoral policies (in this case, subsidies provided in all sectors of the economy) to the overarching aim of sustainable development.

The paper has not taken for granted, that environmental protection and CE are "the same thing" or share the same policy objectives. That's the reason why the paper also had the purpose of making a comparison between the environmental and the CE perspectives in evaluating subsidies, to identify areas of possible contradiction between the two approaches and providing useful information to improve coordination and effectiveness of the two policies.

Given that the CE "schools of thought" are continuously evolving and the CE policy paradigm is quite recent in the EU, it was necessary to carry out a review of the main EU policy documents on the CE (SM 1), preliminary to the policy coherence analysis of subsidies, aimed at clarifying the definition of CE provided by the EU Taxonomy for sustainable finance and deepening the specific concepts and principles to be adopted in the analysis of subsidies, in particular with reference to their effects on land, water and energy resources use and to the conditions for including or excluding the recovery of energy from waste in the CE approach.

The National Catalogue of environmentally harmful and environmentally friendly subsidies, an informative instrument managed by the Ministry of the Environment in collaboration in cooperation with other national administrations, was used as a data set for the analysis. This institutional source, that classifies more than 170 existing subsidies on the basis of a broad taxonomy that includes also tax expenditures and implicit forms of subsidy, enabled a wide qualitative review of the subsidies' coherence with the EU principles of the CE and a simple statistical analysis based on the number of occurrences (number of subsidies of a certain class of coherence, belonging to a certain sector of activity). The method used has precise boundaries and limits; it does not pretend to substitute other

more sophisticated methodologies for assessing subsidies, nor it suggests a specific policy reform of subsidies. It provides to policy-makers an essential type of information, related to: (a) monitoring of subsidies (norms, features, financial size . . . ) and (in the case of this study) (b) assessing their coherence with the CE principles (classification into harmful, friendly or still uncertain subsidies). The major outcomes are a higher transparency and public consciousness of directions where public resources are spent. That is the why the approach is gaining momentum also in the "side-areas" of green budgeting [20,62] and sustainable finance [15,109].

The analysis has made it possible to identify, out of 174 environmentally relevant subsidies in force in Italian legislation (those of the Catalogue), as many as 56 subsidies potentially harmful to the CE (HCE), for a financial value summing up to at least 13.5 billion euros in 2019, and 75 potentially friendly subsidies for the CE (FEC), for at least 13.0 billion euros. Furthermore, in the comparison with the Catalogue result, the subsidies that have had a confirmation of the outcome of the assessment when changing perspective to CE (EHS-HCE, EFS-FCE, EUS-UCE) are 126, equal to 72.4%, while those that have undergone a change of judgement are 43 (about one quarter). Despite the overlapping areas, the environmental protection and the CE approaches also have autonomous aims that can produce contradictory outcomes, jeopardizing the effectiveness of both policies. The results of the study confirm the policy relevance of the CE paradigm but also highlight its inner contradictions and potential inconsistencies with other environmental objectives. There is the need for an environmental approach fully coordinated with the CE perspective: the recent establishment of the Ministry of Ecological Transition through the unification of the Ministry of Environment with a section of the Ministry of Economic Development, is an unique opportunity to assure the needed coordination between CE and environmental protection policies. A practical suggestion for the Italian case is to improve the legal base of the Catalogue by explicitly including the CE perspective in an environmental sustainability one (encompassing all the EU legislation environmental objectives), and to strengthen its robustness in detecting harmful subsidies by a clarification of the technical screening criteria to be used in the assessment (in the spirit of the Delegated acts of the recent EU Taxonomy Regulation). The main recommendation for EU countries is to develop their own "Catalogue of subsidies", since the public availability of a transparent and robust knowledge base on subsidies is a precondition for any environmental fiscal reform, as recognized also by the EU Green Deal.

The analysis can be considered a "pilot study" on a national case, providing a simple method for reviewing existing subsidies with a CE perspective fully integrated with environmental sustainability. The method can be used by competent national administrations also in ex-ante evaluation of new subsidy schemes, and can be easily integrated into current environmental policy assessment practices.

**Supplementary Materials:** The following are available online at https://www.mdpi.com/article/10.3390/su13158150/s1; SM 1—Analysis of circular economy concepts and principles in the main European policy documents on circular economy; SM 2—Tables of Harmful subsidies for the circular economy (HCE) and Friendly subsidies for the circular economy (FCE): Table S1: "Energy sector"—Harmful subsidies for circular economy (HCE); Table S2: "Transport sector"—Harmful subsidies for circular economy (HCE); Table S3: "Other subsidies"—Harmful subsidies for circular economy (HCE); Table S4: "VAT allowances"—Harmful subsidies for circular economy (HCE); Table S5: "Energy sector"—Friendly subsidies for circular economy (FCE; Table S6: "Transport sector"—Friendly subsidies for circular economy (FCE); Table S7: "Agriculture & Fisheries"—Friendly subsidies for circular economy (FCE); Table S8: "Other subsidies"—Friendly subsidies for circular economy (FCE); Table S9: "VAT allowances"—Friendly subsidies for circular economy (FCE).

**Funding:** This research and the APC were funded by the Research Fund for the Italian Electrical System in compliance with the Decree of 16 April 2018.

**Institutional Review Board Statement:** Not applicable.

**Informed Consent Statement:** Not applicable.

**Data Availability Statement:** Not applicable.

**Acknowledgments:** Many thanks to Pierpaolo Girardi (RSE) and Giovanna Martignon (RSE) for their comments on a first and longer version of this research, written in Italian. A slide presentation of a previous version of the paper has been discussed at the workshop "Making the Circular Economy work for Sustainability: From theory to practice", organized by the University of Ferrara, Ferrara & Rovigo, Italy, 23–25 February 2021.

**Conflicts of Interest:** The author declares no conflict of interest.

**Disclaimer:** The views, thoughts and opinions expressed in this paper belong solely to the author, and do not necessarily represent those of the author's employer (RSE).

## Appendix A. CE and Its Concepts: An Historical Overview

CE is a complex, multi-level, socially constructed concept that can be considered a paradigm shift, a toolbox and a conceptual umbrella [10]. In academic literature many variants of the concept are offered, each of them involving partially different principles and continuously evolving approaches [98,110–112]. CIRAIG (International Reference Centre for the Life Cycle of Products, Processes and Services) [110], that attempted an historical timeline of the emergence of the key both academic and policy moments of CE and its associated concepts, places the birth of the term "circular economy" in 1990 (title of Section 2 of the landmark book by D.W. Pierce and R.K Turner, Economics of Natural Resources and the Environment [113]). According to the first EMAF (Ellen MacArthur Foundation) report (2013) [114] the CE concept has deep-rooted origins and cannot be traced back to one single date or author. Its practical applications in business have gained momentum since the late 1970s thanks to the efforts of a small number of academics, thought leaders, and companies. Several "schools of thought" are quoted in EMAF review [114], among which "regenerative Design" [115], "performance economy" [116], "cradle to cradle" [117], "industrial ecology" [118,119]; "biomimicry" [120]. According to EMAF, the common and distinctive element of these approaches is the need for a transition from a linear model of resource consumption to a circular model. Given the continuous demographic and economic growth and scarcity of natural resources, a circular model is needed to avoid supply disruptions and price volatility that put at risk the stability of the financial and economic system. With these premises, the first EMAF report [114] proposed the following definition of CE: "A Circular economy is an industrial system that is restorative or regenerative by intention and design. It replaces the 'end-of-life' concept with restoration, shifts towards the use of renewable energy, eliminates the use of toxic chemicals, which impair reuse, and aims for the elimination of waste through the superior design of materials, products, systems, and, within this, business models" [114] (p. 7).

Two main features can be highlighted in this definition: (a) emphasis is given to elimination of waste through restoration (a broader concept than recycling, that encompasses reuse); (b) efficiency in the use of non-renewable resources is not mentioned.

As to the first point, Korhonen et al. [98] highlight that the restorative nature of the CE approach embeds a hierarchy of principles: at the top of the list there is product reuse, then refurbishment, repair, upgrading, remanufacturing, and only later comes recycling for raw material utilization; recovery of energy from waste use is the second to last option while landfill is the last option. The quality and value of products and their materials should be kept within the economic system as long as possible throughout a "cradle to cradle" life cycle. The concept of economic value maintenance is at the base of the first EU CE action plan of 2015 [34] and, more recently, it's at the heart of the CE definition recently provided by the EU Regulation 2020/852 [15] on the establishment of a framework to facilitate sustainable investment (so called "Taxonomy of environmentally sustainable economic activities"), that is adopted in this paper: "Circular economy means an economic system whereby the value of products, materials and other resources in the economy is maintained for as long as possible, enhancing their efficient use in production and consumption, thereby reducing the environmental impact of their use, minimizing waste

and the release of hazardous substances at all stages of their life cycle, including through the application of the waste hierarchy" [15] (art.2.9).

As compared to EMAF definition [114], it can be noticed the inclusion in the EU definition of CE of the concept of resource efficiency. According to the IRP (International Resource Panel) [121] the term "resource efficiency" encompasses a number of ideas: the technical efficiency of resource use (measured by the useful energy or material output per unit of energy or material input); the resource productivity, or extent to which economic value is added to a given quantity of resources (measured by useful output or value added per unit of resource input); the extent to which resource extraction or use has negative impacts on the environment (impact per unit of resource input). The concept of resource productivity, and its inverse that is resource intensity (resource use per unit of value added), recall the potential to decouple the use of resources from economic growth. In its historical reconstruction of the CE concepts' roots CIRAIG [110] highlights the "unheard message" of the 1972 Club of Rome report on "Limits to growth" [122] and traces the birth of the "sustainable production and consumption" international policy discussion twenty years later (a period during which environmental policies in developed countries were built and consolidated [33,123]), with the Rio 1992 UN Conference on Environment and Development (see Section 4 of Agenda 21) [1] The concepts related to "sustainable production and consumption" have been further developed by the following World Summits on Sustainable development: this international process was fundamental to increase the policy awareness on the urgency of decoupling natural resource use from economic growth [110].

China Government was the first to see in CE an alternative development model [110]: after a first pilot phase on industrial ecology parks, in 2008 adopted the first CE legislation [124]. In the EU, the CE policy process has been much slower. The word "circular economy" appears in the chapter "minerals and metals" of the 2011 Roadmap to a resource efficient Europe [125] (p. 13), that followed the 2005 Thematic Strategy on the sustainable use of natural resources [126] where CE is not mentioned. The start of a CE policy in the EU is marked by the 2015 Action Plan for a CE [34] and the subsequent CE package of legislative proposals on waste [127].

In the same year, the UN 2030 Agenda for sustainable development [2] has included at least three CE-related targets into Goal 12 ("Ensure sustainable consumption and production patterns"): 12.2 "By 2030 achieve the sustainable management and efficient use of natural resources"; 12.3 "By 2030, halve per capita global food waste at the retail and consumer levels and reduce food losses along production and supply chains, including post-harvest losses" and 12.5 "By 2030, substantially reduce waste generation through prevention, reduction, recycling and reuse".

Besides UN, EU and China, CE is currently promoted by many other national governments including Japan, UK, France, Canada, The Netherlands, Sweden, Finland [98] and partly also Italy (with the exception of a preparatory document [70], Italy doesn't have a national plan on CE).

In conclusion, even if the CE "thinking" roots can be found in the seventies, as a policy paradigm it represents a novelty as compared to the much older affirmation of sectoral environmental protection policies in developed countries [33,123].

The delayed policy success of the CE concepts as compared to the times of its founding fathers can be explained by a mix of factors, among which the continuously evolving nature of the paradigm [112], the high variability of policy signals and incompleteness of targets alongside the waste hierarchy of CE [24], the coexistence of too many competing objectives in sustainable development policies [25], the lack of research on policy instruments for the CE [26,32]. A more substantial explanation is probably the late policy acknowledgment of the potential contribute of the CE approach in supplying the huge flux of resource needed by future decarbonized economies through much increased rates and higher quality standards of recycling processes [10,11,14].

## Appendix B. Description of the Data-Set: Summary of the Catalogue Results

In the third edition of the Catalogue, a total of 171 subsidies are analyzed, of which 64 direct and 107 indirect subsidies, for a total financial value of 44 billion euros. The text underlines that this is an underestimate, as many subsidies, albeit classified by the environmental point of view, have not been quantified from the financial point of view (this is a limit that should be overcome in future editions). The Catalogue allocate subsidies to five classes: "Energy" (48 measures), "Agriculture & Fishing" (44 measures), "Other subsidies "(41 measures), "VAT allowances" (22 measures) and Transport (16 measures) (see second-last column of Table A1). This partition uses a mix of not clearly stated criteria, partly sectors of activity, partly tax instruments. For example, subsidies on energy products are found in all classes, except in "Agriculture and fishing".

In financial terms, the Energy sector remains predominant, accounting for 57% of the total amount of subsidies examined with 24.9 billion euros, followed by Agriculture and fishing with 6.6 billion euros (15%).

Coming to the environmental classification results of the Catalogue, 73 subsidies are classified as EFS, 72 as EHS and 26 as EUS (Table A1). Regarding EUS (uncertain subsidies), it is the "Agriculture & Fisheries" sector that records the highest number (13, equal to 50%), followed by the "Other subsidies" sector (7 subsidies).

EFS are concentrated in the "Energy" sector, with 11 mechanism (11.6 billion euro or 76% of EFS—and this is an underestimate, as some mechanisms do not have still been quantified), while "Agriculture & Fisheries" with 23 subsidies benefits of 1.2 billion euro (8% of EFS).

Regarding EHS, in the "Energy" sector 33 subsidies have been found in 2018, for a total of 13.2 billion euros (67% of EHS). The Energy sector is followed by "VAT concessions" (17 EHS, for just over 4 billion euros euro). Worth to note that 83% of EHS and 91% of EHS financial value are related to indirect subsidies (they are mostly exemptions or reductions in excise duties on energy products and "VAT concessions", that are by definition indirect subsidies), while in the "Agriculture & Fishing" sector all EHS are direct subsidies (8 subsidies).

**Table A1.** EFS, EHS and EUS: number of subsidies and financial value, 2018 (million Euro). Source: [19] (tabb. 3.16 and 3.17).

| Sectors | EFS Friendly | | EHS Harmful | | EUS Uncertain | | Total | |
|---|---|---|---|---|---|---|---|---|
| | n. | M€ | n. | M€ | n. | M€ | n. | M€ |
| Energy | | | | | | | | |
| Indirect subsidies | 4 | 77 | 30 | 11.761 | 3 | 78 | 37 | 11.916 |
| Direct subsidies | 7 | 11.568 | 3 | 1.402 | 1 | - | 11 | 12.970 |
| Energy total | 11 | 11.645 | 33 | 13.163 | 4 | 78 | 48 | 24.886 |
| Transport | | | | | | | | |
| Indirect subsidies | 5 | 15 | 4 | 1.637 | | | 9 | 1.651 |
| Direct subsidies | 6 | 24 | | | 1 | 49 | 7 | 73 |
| Transport total | 11 | 39 | 4 | 1.637 | 1 | 49 | 16 | 1.724 |
| Agriculture and fishing | | | | | | | | |
| Indirect subsidies | 2 | 4 | - | - | 2 | 311 | 4 | 315 |
| Direct subsidies | 21 | 1.213 | 8 | 270 | 11 | 4.829 | 40 | 6.312 |
| Agriculture and fishing total | 23 | 1.217 | 8 | 270 | 13 | 5.141 | 44 | 6.627 |
| Other subsidies | | | | | | | | |
| Indirect subsidies | 21 | 2.387 | 9 | 655 | 5 | 1.561 | 35 | 4.604 |
| Direct subsidies | 3 | 6 | 1 | - | 2 | 405 | 6 | 411 |
| Other subsidies total | 24 | 2.393 | 10 | 655 | 7 | 1.966 | 41 | 5.014 |
| VAT allowances | | | | | | | | |
| Indirect subsidies | 4 | 13 | 17 | 4.024 | 1 | 1.416 | 22 | 5.452 |
| VAT allowances total | 4 | 13 | 17 | 4.024 | 1 | 1.416 | 22 | 5.452 |

**Table A1.** *Cont.*

| Sectors | EFS Friendly | | EHS Harmful | | EUS Uncertain | | Total | |
|---|---|---|---|---|---|---|---|---|
| | n. | M€ | n. | M€ | n. | M€ | n. | M€ |
| All sectors | | | | | | | | |
| Indirect subsidies | 36 | 2.495 | 60 | 18.077 | 11 | 3.367 | 107 | 23.938 |
| Direct subsidies | 37 | 12.811 | 12 | 1.672 | 15 | 5.283 | 64 | 19.766 |
| All sectors total | 73 | 15.306 | 72 | 19.748 | 26 | 8.650 | 171 | 43.704 |

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
