# Peer review of "Circular Economy and Environmental Sustainability: A Policy Coherence Analysis of Current Italian Subsidies"

_sustainability, doi:10.3390/su13158150_

Round 1

Reviewer 1 Report

The purpose of the paper is to assess the coherence with the EU recognized circular economy principles of a large set of subsidies currently in force in Italy: those listed in the Italian Catalogue of Environmentally Harmful Subsidies and Environmentally Friendly Subsidies. The analysis allows to identify as many as 56 potentially harmful 13 subsidies for the circular economy in Italy, for a financial value of at least 13.5 billion euros in 2019, 14 and 75 potentially friendly subsidies for the circular economy, for at least 13.0 billion euros.

Introduction

I find the introduction very confusing.

  1. The paper addresses issue of environmentally harmful subsidies devoid of explaining what this could mean - and out of nowhere, the concept of circular economy was introduced without explaining what circular economy means and its interplay or relationship with environmentally harmful subsidies.
  2. The problem statement and objectives of the paper are not clear. Thus, making the arguments in the introduction very difficult to follow.
  3. Not sure about the harmful environmental subsidies this paper aims to focus on, but it is the 'phasing out fossil fuel subsidies, then I will recommend the author to focus on that. Now, the question is, how does this relate to circular economy and under what context?
  4. Which area of EU circular economy is this paper aiming to address?
  5. I am not sure whether line 82-102 is what the study aims to address. If that's the case, then the author must review circular economy versus environmental subsidies literature in the early part of the introduction to strengthen the paper.

Materials

  1. Since I am not sure of what circular economy means in this paper and the direction of the objectives, it isn't easy to gauge what the methodology aims to address.
  2. I find this whole methodology section confusing. What do you mean by "In order to assess the coherence of subsidies with circular economy, a comprehensive and possibly an official source of circular economy principles is needed"? I could see that until you identify the problem statement for this paper, it is too early to make such a statement, as a reader may not know what circular economy is about and under which context.
  3. If the paper is centred on this statement "For the purpose of this paper, the circular economy principles established by art. 13 250 of the EU Regulation 2020/852 [29

Results

The results could mean nothing until you review the introduction and materials, and methods sections. 

Discussion and Conclusion

Still not convinced how the discussion reflects the results and objectives of the study. 

Since the circular economy context is not well explained, including the lack of a total review of EU circular economy policies, some statements made in this paper makes it illegitimate. For example, the article refuses to review 'Circular Economy Perspectives for the Management of Batteries used in Electric Vehicles', yet the author claims analysis of coherence with the EU circular economy principles of a large set of environmentally significant subsidies.

I will also recommend the authors review the whole manuscript and if they may have to consider the issue of environmentally harmful subsidies without implicating it with circular economy.

Reviewer 2 Report

This is a policy relevant paper since its findings are essential in undertaking a green budgetary reform. The question of the budget reform could be placed within the abstract. The author/s have a very good command of the topic presented and results are exhaustive. The results of the paper are quite relevant. The author/s could be more ambitious and beyond the utility of the method presented and its relevance for detecting contradictions, the results should be regarded an illustration of the limits of circular economy (CE).

However, the paper lacks some fundamentals of an academic paper. The most salient is the fact that the author/s do not place the discussion of the topic in relation to previous analysis. This is explicitly clear when looking at the references where very few academic papers are cited. So, one main recomendation is to put this paper in dialogue with other previous academic works. In my opinion, the paper relates to the two following questions: : environmentally harmful subsidies and contradictions of Green Economy.

Therefore, these two matters should be introduced in the text. Just to put some references: 

  1. Environmentally Harmful Subsidies and their reform: Frans H. Oosterhuis, Patrick ten Brin (eds).2014. Paying the Polluter: Environmentally Harmful Subsidies and their Reform;  Phasing out public financial flows to fossil fuel production in Europe https://doi.org/10.1080/14693062.2020.1736978 ; Reforming fossil fuel subsidies: drivers, barriers and the state of progress https://doi.org/10.1080/14693062.2016.1169393;
  2. Contradictions of the CE. Thermodynamic Rarity and Recyclability of Raw Materials in the Energy Transition: The Need for an In-Spiral Economy https://doi.org/10.3390/e21090873

The author/s talk about CE principles, though CE is not explained, neither CE principles developed. In my opinion, this should be introdeced in the article. See for instance: Circular Economy: The Concept and its Limitations https://doi.org/10.1016/j.ecolecon.2017.06.041

Another question that requires some further development is the critique (and analysis of contradictions) of the official statements. The paper is grounded on policy reports and official guidelines, but those might be contradictory. Those guidelines are the ones orienting the definitions used by the author/s: EHS, EFS, EUS, HCE, FCE, UCE, NCE. An example that illustrates this is when the author/s state that: "The positive motivation of the Catalogue is based on the fact that Directive 2014/94 [26] includes LPG among the "alternative fuels" to be promoted in the EU and on the lower harmful emissions of LPG as compared to other fuels used in such areas.”

In the conclusions and discussion section, the author/s could highlight not only the contradiction detected in subsidies that might be friendly from a CE standpoint, but harmful from an environmental approach; but develop some of the problematics related to the very same official definition of CE. Therefore, official CE guidelines and principles should be reviewed.

Reviewer 3 Report

Please see the attached file for details of my comments.

Reviewer 4 Report

The article aims to examine the coherence between Circular Economy (CE) principles, as recognized by the European Commission, and a large set of subsidies listed in the Italian Catalogue of Environmentally Harmful Subsidies and Environmentally Friendly Subsidies. The author seeks to analyse the policy coherence to see if Italian subsidies meet CE objectives in addition to environmental objectives, whether subsidies are overlapping in the objectives if they are harming or contributing to achieving respective objectives. While the original contribution is well explained and demonstrated, there are major amendments to be made to improve the quality and clarity of the research namely in the structure and length of the article especially in the first parts of the paper. Therefore, I would recommend the following major revisions and amendments to enhance the quality of the paper.

Broad comments

  • The Introduction section is too long. An introduction should never be 11 pages. I would highly suggest creating a literature review section (as a section 2) made up of the current 1.1 What is circular economy and its basic concepts, 1.2. The debate on strengths and weakness of CE in an environmental sustainability framework, and 1.3 Environmentally harmful subsidies and the “missing debate” on alignment of subsidies with circular economy.
  • The introduction should mention all the main important elements of the study. I would therefore highly recommend moving section 1.4 Aim of the study to the first part of the introduction on Page 2 and reducing the “Aim of the study” section. It is important to present in the first pages of the article the aim of the paper and explain the reasons why the author has chosen Italy as a case study for example.
  • I would also recommend adding some indication regarding the methods in the introduction. Mentioning that the study is fulfilling the aim by conducting a review of policy documents on CE and a Policy Coherence for Sustainable Development (PCSD) method based on logic rules developed by the OECD.
  • In general, the sections of the paper are quite long, with the literature review section being 9 pages and the Materials and Methods section being 8 pages. The reader might be losing the thread of the story and thus the main point of the study when there are too many indirect contextualizations and details that are not necessary considering the scope of the study. There is potential for each section to be more succinct and straightforward. Therefore, I would recommend the author to read through the article and “declutter” the unnecessary details and information (especially in 1.3 and 2) and evaluate whether the information contributes to the presentation, explanation, and interpretation of the results. For example, I would recommend deleting some of the very specific historical and anecdotal information about the global context and deleting details for instance about Germany and France (page 9, lines 414-436) to be able to move more quickly to the national level and Italian context which is the focus of the paper.
  • Please define and present the acronyms the first time they appear and consistently apply the acronyms thereafter. For instance, explain Circular Economy (CE) principles on page 1, line 33 (the first line) instead of on page 2, line 77 when it has already been mentioned several times. Also, explain for instance, “CIRAIG” (page 2, line 85), “EMAF” (page 2, line 89) or “International Monetary Fund” (page 6, lines 293-294), or “SM 1” (page 8, line 368).
  • I would recommend going through the paper and making sure capital letters are applied where appropriate, for instance “International Rresource Panel [25]” (page 3, line 121).
  • I would recommend shortening the headings. For instance, “1.1 Circular Economy and its basic concepts”, “1.2. Strengths and weakness of CE” and so on.
  • I would recommend systematically present numbers numerically when appropriate thorough the paper. For instance, “…the Catalogue assessed more than one hundred and seventy 170 subsidies from the environmental point of view” (page 11, lines 531-532).

Specific comments per section 

  1. Introduction
  • Page 2, lines 48-52: I would recommend contextualizing some of the quotes. Could be as simple as X stated in this report that "..."
  • Page 2, line 58 and 64: Please do not put a capital letter to “country”, “countries” or “nation”. Capital letters only apply to the names of specific countries.
  • Page 2, lines 59-62: “As will soon be explained, the issue of the policy coherence of subsidies has been much more debated 60 in relation to environmental objectives (are a nation’s subsidies harmful or favorable for the environment?), both by researchers and policy makers, at least for fifteen years.” Please add references to this sentence. Who are the main researchers and policy makers that have been debating for 15 years?
  • Page 2, lines 63-66: as reducing air pollution, mitigating climate change, reducing polluted liquid discharges, waste prevention, recycling and disposal, protecting natural ecosystems, slowing and preventing biodiversity loss, use of fishing sources within their renovation rates…”. The list of environmental objectives could be reduced to be more concise.
  • Page 2, line 68: “Contradictions between the two perspectives in the design of a 68 subsidy can be found;” Please add references to this sentence.
  • Page 2, lines 76-80: Please present the structure of the rest of the whole paper and not just the next few sections.

1.1 What is circular economy and its basic concepts

  • Page 3, lines 99-104: Please provide some comment on the cited definition.
  • Page 3, lines 120.121: From the above definition it can be noticed the inclusion of another pillar of CE, that is the concept of resource efficiency.” From which definition exactly? The author provides two different definitions above.
  • I would suggest presenting only one definition of CE which would be the one that is adopted for the study.

1.2. The debate on strengths and weakness of CE in an environmental sustainability framework

  • Page 4, lines 190-193: The sentence is unclear. Please simplify it. Does the author mean that the Taxonomy Regulation has not included CE in the range of environmental objectives?
  • Page 4, line 199: Please delete “Not less important”
  • Page 5, lines 219-220: (the example provided is the use of biomasses imported from countries where they are produced inducing land use change and loss of biodiversity).” Where is the example provided? This part of the sentence could be deleted.

1.4. Aim of the study

  • Page 10, line 462: aimed to detecting harmful of or friendly subsidies”.
  • Page 10, line 472: I would recommend deleting the sentence: In the EU context at least three States have voluntarily developed a report on subsidies [56, 57, 80]”
  1. Material and Methods

2.1 The data set used: the Catalogue of environmentally harmful and of environmentally friendly subsidies

  • Page 11, lines 524-531: The sentence is too long. I would recommend rephrasing and shortening the sentence.

2.2. Methodology

  • Page 15, lines 685-686: The main conclusions are 685 summarized below”. Please specify: the main conclusion of what exactly?
  • Page 16-17, lines 744-766: Again, the Methodology section on its own is long (3 pages). I would recommend deleting the details of the objectives and the context of how the method came about (lines 744-766). The paper would really gain clarity and show its original contribution if the main thread of the story is focused on the aim of the paper. For the methods section, the readers should be able to quickly know which method was used and how it was used for the study.
  1. Results
  • The results section is well structured, clear, and easy to follow. Very interesting to read!
  • Page 20, line 894: Please explain the acronym LPG. 
  1. Discussion and conclusions

4.1. Discussion

  • I would suggest renaming section 4 to a more specific title such as “Methodological and Analysis Limitations”. This would help for clarity purposes by already ready the heading, the readers know the focus of the section, of the “discussion”.

4.2. Conclusions

  • The conclusions section is great, clear, and concise. It could be a model or example of how the other sections can be improved.
  • I would recommend having the conclusions as a chapter on its own, as chapter 5.

My best wishes with this paper.

Reviewer 5 Report

The article concerns an important and current issue about the circular economy and subsidy. The theoretical introduction is very detailed but too long in my opinion. The weak point is the article is the presentation of the approach of only one country (especially after such a rich and extensive theoretical introduction)

Round 2

Reviewer 3 Report

Although the paper had some improvements, I think the overall contents are still far from an academic paper. Furthermore, the paper is very lengthy and should consider removing repetitive sentences.

  1. I think the introduction section is too long. Sections 1.1-1.3 should be separated into section 2 and the introduction should focus more on a simple explanation of the motivation and significance of the research objectives.
  2. I did not see any relevant studies using a similar policy coherence method. For example, Nilsson et al (2012) conduct a study using a similar method. https://doi.org/10.1002/eet.1589
  3. Similar to the way the methods are presented in Nilsson et al., some graphical summary of the methods would make it easier to understand the step of the methods.
  4. I think the conclusion should be more focused on the policy implications driven from the results rather than including the summary and limitations of the study.

Reviewer 4 Report

The author has managed to address all the comments appropriately and clearly. The article’s structure and content are now much clearer and well organized, and the length is now appropriate. Therefore, I would support publishing with minor revisions left to be made namely regarding some grammar issues and typos:

  • I would recommend proof-reading the paper to identify grammar and spelling mistakes especially the use of prepositions such as in Page 1, line 34: deal about (--> deal with) or Page 2, line 51: coupled by (--> coupled with).
  • I would recommend changing the word order and adding the reference corresponding the “wasteful resource practices” expression on page 2, line 81.
  • I would recommend going through the paper and again making sure capital letters are applied or deleted where appropriate, for instance “To coherently align a Nnation’s subsidies with the environmental and CE objectives under the auspices of the UN policy framework of the 2030 Agenda for sustainable development [2]” (page 3, line 98).
  • Please check that the acronyms are written correctly (for example UE page 2, line 59) and explained such as IEEP page 4, line 411. 

My best wishes with this paper.

Round 3

Reviewer 3 Report

Overall comments

I think the authors have mostly met the comments suggested in the previous comments. However, I feel the paper is a little too long so I recommend shortening it, Furthermore, there are still some minor issues to be addressed.

Specific comments

  1. It seems that “policy coherence analysis” (PCA) is a more commonly used term than the simple coherence analysis (CA) so I recommend replacing CA with PCA.
  2. Ln50, UE should be EU.
  3. Explanations on why the categories 4.1.1 to 4.1.5 were analyzed should be provided in the methods section.
  4. Ln856-858. Avoid including URLs in the main text. It should be included in the reference.
